# Deep Learning Through the Lens of Example Difficulty

**Robert J. N. Baldock**[*]
Google Research, Brain Team
rjnbaldock@gmail.com

**Hartmut Maennel**
Google Research, Brain Team
hartmutm@google.com

**Behnam Neyshabur**
Google Research, Blueshift Team
neyshabur@google.com

## Abstract

Existing work on understanding deep learning often employs measures that compress all data-dependent information into a few numbers. In this work, we adopt a perspective based on the role of individual examples. We introduce a measure of the computational difficulty of making a prediction for a given input: the *(effective) prediction depth*. Our extensive investigation reveals surprising yet simple relationships between the prediction depth of a given input and the model's uncertainty, confidence, accuracy and speed of learning for that data point. We further categorize difficult examples into three interpretable groups, demonstrate how these groups are processed differently inside deep models and showcase how this understanding allows us to improve prediction accuracy. Insights from our study lead to a coherent view of a number of separately reported phenomena in the literature: early layers generalize while later layers memorize; early layers converge faster and networks learn easy data and simple functions first.

## 1 Introduction

Much of the existing work on understanding deep learning "integrates out" the data, viewing the inductive bias of the model, or the properties of the optimizer as central to the success of the approach. Examples of such work include studies of eigenvalues of the Hessian and the geometry of the loss landscape (Ghorbani et al., 2019; Yao et al., 2020; Sagun et al., 2016; Li et al., 2018; Pennington and Bahri, 2017; Sagun et al., 2018), studies of margin and effective generalization measures (Long and Sedghi, 2019; Unterthiner et al., 2020; Jiang et al., 2020, 2018; Kawaguchi et al., 2017) and mean-field studies of stochastic optimization (Smith et al., 2021; Stephan et al., 2017; Smith and Le, 2018). However, in practice, we are rarely concerned with only the average behavior of a model.

One pathway to understanding the principles that govern how deep models process data is to study the properties of deep models for data points with different "amounts" or "types" of *example difficulty*. There are a number of definitions of example difficulty in the literature (E.g. see Carlini et al. (2019); Hooker et al. (2019); Lalor et al. (2018); Agarwal and Hooker (2020)). Two are particularly relevant to this work. Firstly, the probability of predicting the ground truth label for an example, when that example is omitted from the training set (Jiang et al., 2021), which represents a *statistical view* of example difficulty. Secondly, the difficulty of learning an example, parameterized by the earliest training iteration after which the model predicts the ground truth class for that example in all subsequent iterations (Toneva et al., 2019). This measure represents a *learning view* of example difficulty [2].

---

[*]Work completed as part of the Google AI Residency Program

[2]We expand on other notions of example difficulty in Appendix B.

35th Conference on Neural Information Processing Systems (NeurIPS 2021).

These notions suffer from two fundamental limitations. While early-exit strategies in computer vision (Teerapittayanon et al., 2016; Huang et al., 2018) and NLP (Dehghani et al., 2018; Liu et al., 2020b; Schwartz et al., 2020; Xin et al., 2020) suggest predictions for easier examples require less computation, the above example difficulty notions do not encapsulate the processing of data inside a given converged model. Moreover, existing notions of example difficulty (E.g. Carlini et al. (2019)) provide a one-dimensional view of difficulty which can not distinguish between examples that are difficult for different reasons.

In this paper, we take a significant step towards resolving the above shortcomings. To take the processing of the data into account we propose a new measure of example difficulty, the prediction depth, which is determined from the hidden embeddings. To escape the one-dimensional view of difficulty, we introduce three distinct difficulty types by relating the hidden embeddings for an input to high-level concepts about example difficulty: "Does this example look mislabeled?"; "Is classifying this example only easy if the label is given?"; "Is this example ambiguous both with and without its label?". Furthermore, we show how this enhanced notion of example difficulty can unify our understanding of several seemingly unrelated phenomena in deep learning. We hope that the results presented in this work will aid the development of models that capture heteroscedastic uncertainty, our understanding of how deep networks respond to distributional shift, and the advancement of curriculum learning approaches and machine learning fairness. These connections are discussed in Section 5.

**Contributions**    Our main contributions are as follows:

- We introduce a measure of *computational example difficulty*: the *prediction depth* (PD). The prediction depth, illustrated in Figure 1, represents the number of hidden layers after which the network's final prediction is already (effectively) determined (Section 2).

- We show that the prediction depth is larger for examples that visually appear to be more difficult, and that prediction depth is consistent between architectures and random seeds (Section 2.2).

- Our empirical investigation reveals that prediction depth appears to establish a *linear* lower bound on the consistency of a prediction. We further show that predictions are on average more accurate for validation points with small prediction depths (Section 3.1).

- We demonstrate that final predictions for data points that converge earlier during training are typically determined in earlier layers which establishes a correspondence between the training history of the network and the processing of data in the hidden layers (Section 3.2).

- We show that both the adversarial input margin and the output margin are larger for examples with smaller prediction depths. We further design an intervention to reduce the output margin of a network and show that this leads to predictions being made only in the latest hidden layers (Section 3.3).

- We identify three extreme forms of example difficulty by considering the prediction depth in the training and validation splits independently and demonstrate how a simple algorithm that uses the hidden embeddings in one middle layer to make predictions can lead to dramatic improvements in accuracy for inputs that strongly exhibit a specific form of example difficulty (Section 4).

- We use our results to present a coherent picture of deep learning that unifies four seemingly unrelated deep learning phenomena: early layers generalize while later layers memorize; networks converge from input layer towards output layer; easy examples are learned first and networks present simpler functions earlier in training (Section 5).

**Experimental Setup:** To ensure that our results are robust to the choice of architectures and datasets, we report empirical findings for ResNet18 (He et al., 2016), VGG16 (Simonyan and Zisserman, 2015) and MLP architectures trained on CIFAR10, CIFAR100 (Krizhevsky et al., 2009), Fashion MNIST (FMNIST) (Xiao et al., 2017) and SVHN (Netzer et al., 2011) datasets. All models were trained using SGD with momentum. Our MLP comprises 7 hidden layers of width 2048 with ReLU activations. Details of the datasets, architectures, and hyperparameters used can be found in Appendix A.

**Related Work:** Our work uses hidden layer probes to determine example difficulty. We have discussed how our study relates to prior work on example difficulty. Hidden layer probes have also been used to study deep learning. Deep k-NN methods (Papernot and McDaniel, 2018) determine their predictions and estimate their own uncertainties by comparing the hidden embeddings of an input to those of the training set. Cohen et al. (2018) showed that SVM, k-Nearest Neighbors (k-NN)

and logistic regression probes achieve similar accuracies. However, they did not study the processing of individual data points nor did they relate the k-NN accuracy to notions of example difficulty. Alain and Bengio (2017) used linear classifier probes in the hidden layers to interrogate deep models and demonstrated that linear separability of the embeddings increases monotonically with depth. We provide a more detailed discussion of related work in Appendix B.

## 2 Prediction Depth: a Computational View of Example Difficulty

We discussed the *statistical* and *learning* views of example difficulty in Section 1. In this section, we introduce a *computational* view of example difficulty parametrized by the prediction depth as defined in Section 2.1. This computational view asserts that, for "easy" examples, a deep model's final prediction is effectively made after only a few layers, while more layers are used for "difficult" examples.

### 2.1 Definition

Asserting that the final prediction is effectively determined in earlier layers of a model, before the output, we estimate the depth at which a prediction is made for a given input as follows [3]:

1. We construct k-NN classifier probes from the embeddings of the training set after particular layers of the network, including the input and the final softmax. The placement of k-NN probes is described in Appendix A.5. We use $k = 30$ in the k-NN probes. Appendix A.4 establishes that the k-NN accuracies we report are insensitive to $k$ over a wide range.

2. A prediction is defined to be made at a depth $L = l$ if the k-NN classification after layer $L = l - 1$ is different from the network's final classification, but the classifications of k-NN probes after every layer $L \geq l$ are all equal to the final classification of the network. Data points consistently classified by all k-NN probes are determined to be (effectively) predicted in layer 0 (the input) [4].

It is worth noting that the prediction depth can be calculated for all data points: both in the training and validation splits. This leads to two notions of computational difficulty:

- The difficulty of predicting the (given) class for an input (in the training split)
- The difficulty of making a prediction for an input, unseen in advance (from the validation split)

We examine both notions of computational difficulty in this paper and use the distinction between them to describe different forms of example difficulty in Section 4.

### 2.2 Prediction depth is a meaningful and robust notion of example difficulty

In this section we show that prediction depth agrees with intuitive notions of example difficulty and that it is consistent between different training runs and similar architectures.

**Prediction depth is higher for examples and datasets that seem more difficult**  If prediction depth is a sensible measure of example difficulty then we would expect the following sanity checks to be observed:

1. Individual data points that are visually confusing or mislabeled should have larger prediction depths as compared to images that are clear examples of their class.
2. Data points from tasks that are intuitively simpler should have lower prediction depths on average.

Figure 1 shows that the prediction depth passes both of these sanity checks. Appendix C.1 presents additional images, providing further evidence for this claim.

**Prediction depth is consistent across random seeds and similar architectures**  Figure 2 shows that the prediction depth is highly consistent between different architectures and random seeds for all datasets. Perfect agreement is not expected as different deep learning algorithms have different inductive biases which affects the perceived difficulty of examples. We observe stronger correlation

---

[3]In the process of arriving at this definition of the prediction depth we considered several alternatives, including using the ground truth class in place of the predicted class and using logistic regression probes in place of k-NN probes. See Appendix E for a discussion on the choices we made in our definition.

[4]Implementation details can be found in Appendix A.6.1.

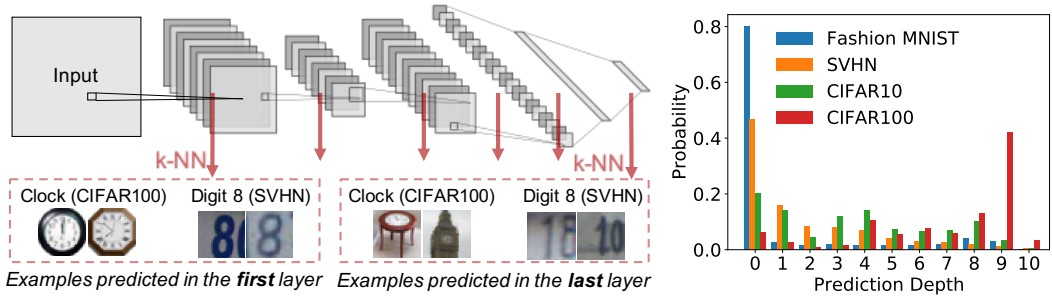

Figure 1: *Deep models use fewer layers to (effectively) determine the prediction for easy examples and more layers for hard examples.* **Left:** A cartoon illustrating the definition of prediction depth (given in Section 2.1). Also shown are training examples from CIFAR100 ("Clock") and SVHN ("Digit 8"). The examples shown are predicted at the input (first layer) or softmax (last layer) of ResNet18. The examples predicted in the input are visually typical ("easy"), while those predicted in the softmax are mislabeled and/or visually confusing ("hard" examples). To find the prediction depth, we build k-NN classifiers from the embeddings of the training set in different layers of the model. The prediction depth corresponds to the earliest layer at which the predictions of all subsequent k-NN classifiers converge to a fixed label. **Right:** Probability of prediction depth in ResNet18 models for four datasets (training split). We see that the four distributions have different characteristic prediction depths. Ranking the mean prediction depths of these datasets in ascending order, we observe: Fashion MNIST (smallest), SVHN (second), CIFAR10 (third), and CIFAR100 (largest). This order aligns with how one might intuitively rank the difficulties of these classification tasks.

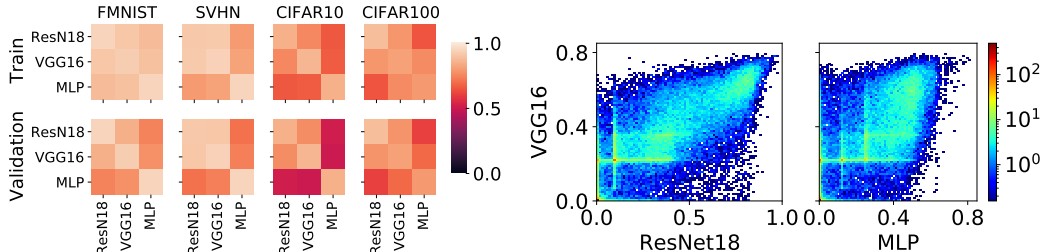

Figure 2: *Consistency of prediction depth between architectures and random seeds.* **Left:** The panel shows the correlation coefficient between prediction depths in different architectures, for both train and validation splits in four datasets. Diagonal comparisons between an architecture and itself show the correlation for the same architecture trained with different random seeds. **Right:** Histograms comparing the mean value of prediction depth obtained for each data point in the training set of CIFAR10 from an ensemble of 250 trained models. In this plot, for visual simplicity, we rescale prediction depth to the interval $[0, 1]$ for each network. Similar results for all other datasets are presented in Appendix C.2.

between prediction depth for ResNet18 and VGG16, than between VGG16 and MLP. This may be explained by the fact that ResNet18 and VGG16 are both convolutional networks and we expect their inductive biases to be more similar to one another than to MLP.

## 3 Deep Learning Phenomena Through the Lens of Prediction Depth

In this section, we explore how the prediction depth can be used to better understand three important aspects of Deep Learning: accuracy and consistency of a prediction; the order in which data is learned and the simplicity of the learned function (as measured by the margin) in the vicinity of a data point.

### 3.1 Depth of a prediction gives a linear lower bound on its consistency

Adopting a *statistical view* of example difficulty, Jiang et al. (2021) identified example difficulty with the expected accuracy of the learning algorithm for a given input, averaged over models trained on different random subsets of the training set with different random seeds. In this section, we clarify the relationship between the prediction depth and the expected accuracy by disentangling the accuracy from the sensitivity of predictions to the particular training split and random seed. Following Jiang et al. (2021), we measure the expected accuracy using the consistency score.

**Consistency score $\hat{C}$:** The frequency of classifying an example correctly when it is omitted from the training set. An empirical estimator of the consistency score for a validation point $(x, y)$

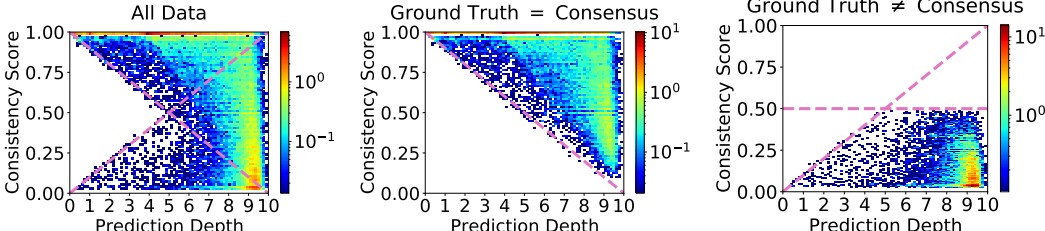

Figure 3: *Consistency score vs. prediction depth in the validation split (left) can be understood as the superposition of two simple functions (middle and right).* We trained 250 ResNet18 models on CIFAR10, with 90:10% random train:validation splits as described in Appendix A. These histograms compare the frequency of correct predictions to the average prediction depth for a data point when it occurs in the validation split. The density of data points is indicated by the color bar, which follows a log scale. The average prediction depth forms two, surprisingly simple, linear bounds on the consistency score (see Section 3.1 for a full description.) This Figure is reproduced for all datasets and architectures in Appendix C.3, illustrating the consistency of this result.

is given by (Jiang et al., 2021):

$$\hat{C}_{A,\mathcal{S}}(x,y) = \hat{\mathbb{E}}^r_{\tilde{\mathcal{S}} \overset{n}{\sim} \mathcal{S} \setminus \{(x,y)\}} [\delta_{y_A,y}] \tag{1}$$

where $A$ is a deep learning algorithm (architecture, loss and optimizer), $y$ is the ground truth class for $x$, $\tilde{\mathcal{S}}$ is a random subset of $n$ points sampled from a training dataset $\mathcal{S}$ excluding $(x, y)$, $y_A$ is the predicted class of $x$ for $A$ trained with data $\tilde{\mathcal{S}}$, $\delta$ is the Kronecker delta and $\hat{\mathbb{E}}^r$ denotes empirical averaging with $r$ i.i.d. samples of such subsets $\tilde{\mathcal{S}}$.

Figure 3 (left panel) shows the relationship between consistency score and prediction depth. This plot indicates a surprising piecewise linear boundary which is symmetric around consistency score $\frac{1}{2}$. This suggests the existence of a missing concept that could simplify the picture. We next show that the missing concept is the notion of a consensus class which is defined below.

**Consensus class $\hat{y}_A$:** The *consensus class* of $x$ is defined as the predicted class for input $x$ by a majority voting ensemble of $r$ models each of which is trained on a randomly chosen subset $\tilde{\mathcal{S}} \overset{n}{\sim} \mathcal{S} \setminus \{(x, y)\}$ [5].

Figure 3 (middle and right) shows how conditioning on whether consensus class matches the ground truth can change the relationship between consistency score and the prediction depth. For points where the consensus class matches the ground truth (middle) we see that the prediction depth forms a, surprisingly simple, linear lower bound on the consistency score. For points where the consensus class differs from the ground truth (right) at low prediction depth the consistency score is bounded from above by a line that reflects the bound from the middle plot in $\hat{C} = \frac{1}{2}$, suggesting that such points are repeatedly mislabeled with a wrong class label. At high prediction depth, the consistency score is low, which suggests highly inconsistent predictions and low accuracy. This result suggests a simple hypothesis: that predictions with low prediction depth are consistent with the *consensus class*, whether that matches the ground truth class or not, while predictions made in later layers depend strongly on the specific training split and random seed used for training and initialization. We measure consistency with the consensus class using the consensus-consistency score.

**Consensus-consistency score $C^*$:** The fraction of models in an ensemble that predict the ensemble's consensus class $\hat{y}_A(x)$ for an unseen input $x$.

$$C^*_{A,\mathcal{S}}(x) = \hat{\mathbb{E}}^r_{\tilde{\mathcal{S}} \overset{n}{\sim} \mathcal{S} \setminus \{(x,y)\}} [\delta_{y_A, \hat{y}_A(x)}] \tag{2}$$

where the notation is the same as in (1) [6].

Figure 4 (left) establishes that our simple hypothesis is indeed correct: the prediction depth forms a linear lower bound on the consensus-consistency score for all data points, irrespective of whether the

---

[5]Implementation details can be found in Appendix A.6.2

[6]Consensus-consistency score is a measure of uncertainty and can be used for calibration (Lakshminarayanan et al., 2017; Wenzel et al., 2020; Wen et al., 2019). See Appendix A.6.3 for details of our implementation.

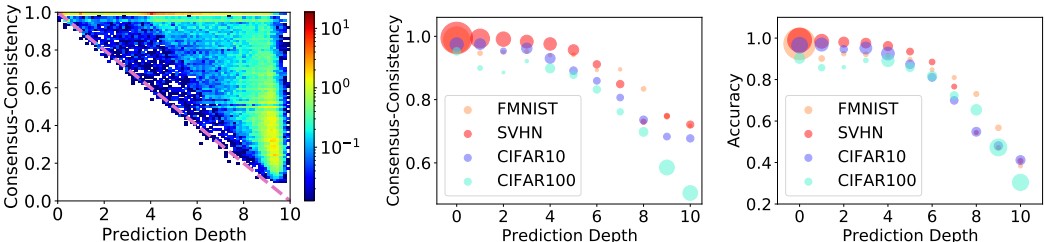

Figure 4: **Left:** *Prediction depth provides us with a linear lower bound on consensus-consistency.* Results for CIFAR100 with ResNet18. We train 250 models (90:10% random train:validation splits) and compare the average prediction depth when a point occurs in the validation set, to the consensus-consistency of the corresponding predictions. Predictions made for points with low mean prediction depths are highly consistent. Conversely, predictions for points with high mean prediction depths are typically more sensitive to the particular training split and random seed used during training. This left plot shows the result for CIFAR100 with ResNet18. The density of data points is indicated by the color bar, which follows a log scale. **Middle:** *Prediction depth in one model predicts the consensus-consistency of an ensemble that does not include that model.* For each dataset we train 25 ResNet18 models with the full training set (see Appendix A). The consensus-consistency of each test point is obtained from 24 of the models, while the prediction depth is obtained from the remaining 1 model. We see that prediction depth in one model predicts the consensus-consistency of a separate ensemble: a measure of the uncertainty of the prediction. The size of each marker in the middle and right plots shows the fraction of the dataset with each prediction depth. Reaffirming the second sanity check in Section 2.2, and in agreement with Figure 1 (right), intuitively simpler datasets (Fashion MNIST and SVHN) have low average prediction depths, while CIFAR100 (intuitively the hardest dataset) has the largest average prediction depth. **Right:** *Prediction depth predicts accuracy.* For each dataset we train 250 ResNet18 models (90:10% random train:validation splits). Each time a point appears in the validation split we record the prediction depth and whether the prediction was correct. Predictions made in earlier layers are more likely to be correct. Consistency of these plots is demonstrated for all datasets and architectures in Appendix C.3 where we also describe the relationship between the prediction depth and the *entropy of the predictions* for an ensemble.

consensus class matches or differs from the ground truth. Interestingly, Figure 4 (middle and right) shows how the prediction depth in a single model, can be used to estimate both of these quantities. That is, predictions of data points with lower prediction depth are both more likely to be consistent and more likely to be correct.

## 3.2 The prediction depth of an input is correlated with its learning difficulty

In Section 3.1, we describe the relationship between the prediction depth, which represents a *computational view* of example difficulty and the consistency and consensus-consistency scores, which represent a *statistical view*. In this section we compare prediction depth to a *learning view* of example difficulty. We measure the difficulty of learning an example by the speed at which the model's prediction converges for that input during training. The following definition is adapted from Toneva et al. (2019):

**Iteration learned** A data point is said to be learned by a classifier at training iteration $t = \tau$ if the predicted class at iteration $t = \tau - 1$ is different from the final prediction of the converged network and the predictions at all iterations $t \geq \tau$ are equal to the final prediction of the converged network. Data points consistently classified after all training steps and at the moment of initialization, are said to be learned in step $t = 0$ [7].

Figure 5 (left plot) shows the positive correlation between the prediction depth and the iteration learned, for all four datasets in VGG16. Consistent results are presented for all architectures and datasets, in both the validation and training splits in Appendix C.4. As a result of the reported correlation, we anticipate that many of the data points correctly classified by the k-NN probe in a particular layer should also be correctly classified by the network at a corresponding interval of training steps. If this is correct then we would expect there to be a visual correspondence between the *training learning curve* (which shows how the accuracy of the network changes during training) and the accuracy of the k-NN probes as data passes from input, through the network, towards the output layer. We call the series of k-NN probe accuracies the *inference learning curve*.

---

[7]Note that this definition can be applied to points in both training and validation splits. In order to compare different models and datasets we rescale the iteration learned in each model so that the 95th percentile occurs at 1.0 and network initialization at 0.

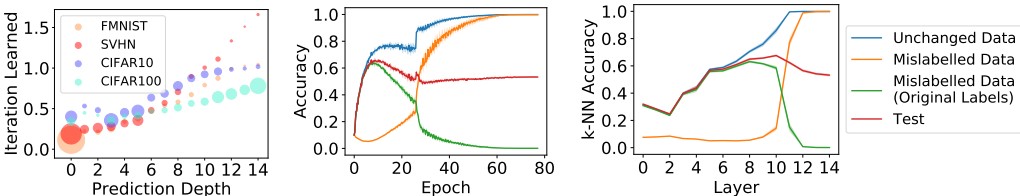

Figure 5: **Left:** *Data points with small prediction depths are on average learned before data points with higher prediction depths.* We train 250 VGG16 models for each dataset, using a 90:10% random train:validation split as described in Appendix A. Each time an input appears in the validation split we record the prediction depth and the iteration learned in that model. This plot shows the average iteration learned for data points at each prediction depth. Marker size shows the fraction of the dataset with each prediction depth. The Pearson correlation coefficients for the four data sets are as follows. CIFAR100: 0.83. CIFAR10: 0.7. Fashion MNIST: 0.79. SVHN: 0.77. **Middle and right:** *The training learning curve (middle) shares several important features with the inference learning curve (right).* Blue, yellow and green curves represent different components of the CIFAR10 training split, in which we have randomized (and fixed) 40% of the labels, and red curves show the test split. The middle and right plots show results from 5 random seeds. The inference learning curve (right) is the sequence of k-NN probe accuracy values for each split. All three plots show results for VGG16. The hyperparameters used are given in Appendix A.

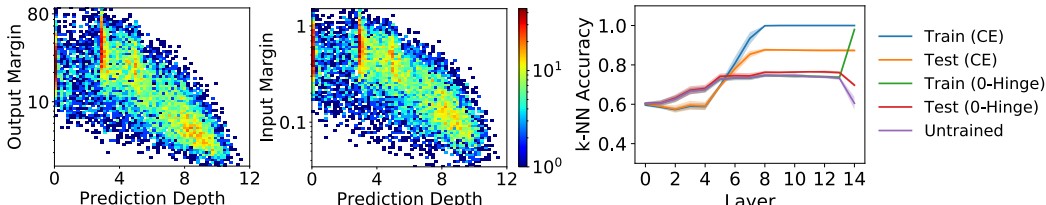

Figure 6: **Left and Middle:** *Test examples with smaller prediction depths, on average, have larger output and input margins.* We train 25 VGG16 models with different random seeds on CIFAR10 (see Appendix A for details) and compare the mean prediction depth of each test point in these 25 runs to its mean output and input margins (log scales). Correlation coefficients are $-0.70$ (output margin) and $-0.69$ (input margin). The density of data points is indicated by the color bar, which follows a log scale. Although the prediction depth could be at most 14, no data point has an average prediction depth greater than 12. **Right:** *An intervention that does not encourage large output margin ("0-Hinge") results, as predicted, in models where the predictions are effectively determined in higher layers in the network compared to the standard training ("CE").*

To test this hypothesis we train a model on a training split where a subset of labels are corrupted and compare the training and inference learning curves on four splits of the data: unchanged training data; mislabeled training data; the original labels of the mislabeled training data and the test split. In Figure 5 (middle and right plots) we see that many of the important features of the training learning curve are indeed present in the inference learning curve. During training (middle), mislabeled data are initially processed as though they are a member of their original class (before they were mislabeled) (Liu et al., 2020a). After an initial period of learning, the network begins to learn the new (random) labels that have been assigned to those data points, so the orange curve moves upwards, and the green curve downwards. At this point, a maximum is observed in the training accuracy (Arpit et al., 2017). In the right plot we see that these same phenomena occur in the inference learning curve.

### 3.3 Deep models exhibit larger margins for inputs with lower prediction depth

It is reported in the literature that deep networks learn functions of increasing complexity during training (Hu et al., 2020; Kalimeris et al., 2019). We frame this observation differently: the learned function is "locally simpler" in the vicinity of data points with smaller prediction depths, and these points are typically learned earlier in training (Section 3.2).

Two known measures of the simplicity of a learned function are the output margin (the difference between the largest and second-largest logits) and the adversarial input margin (the smallest norm required for an adversarial perturbation in the input to change the model's class prediction). We estimate the adversarial input margin, $\gamma$, with a linear approximation (Jiang et al., 2018): for an input $x$ with predicted class $i$, $\gamma \simeq \min_{j \neq i} \frac{|z_i - z_j|}{|\nabla_x(z_i - z_j)|}$ where $z_j$ is the logit returned by the network for class $j$. Figure 6 (left and middle plots) show that data points with smaller prediction depths have

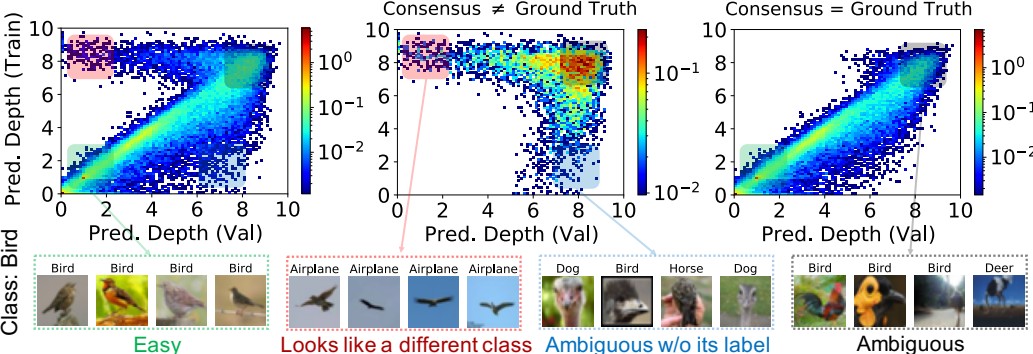

Figure 7: *The prediction depth can be the same, or very different for the same input when it occurs in the train and validation splits. Corners of this plot correspond to different forms of example difficulty.* (See Section 4 for discussion.) We train 250 ResNet18 models on CIFAR10 with random 90:10% train:validation splits as described in Appendix A. These histograms compare average prediction depth for each data point when it occurs in the validation split vs the training split. This behavior is consistently reproduced for all datasets and architectures in Appendix C.6. Below we show extreme (not hand-chosen) images of "Birds" that appear closest to the corners of this plot. The consensus class is given above each image (tiebreaks favor the class "Bird".)

both larger input and output margins on average and that variances of the input and output margins decrease as the prediction depth increases.

To illustrate the strength of the relationship between the prediction depth and output margin, we demonstrate that reducing the output margin of the learned function results in a model that clusters the data only in the latest layers: such a solution has a very high average prediction depth. We do not minimize the output margin directly but rather use a loss and an optimizer that do not encourage high output margin. Naturally there are many unknowns that may contribute to this effect. We simply report the intervention and the outcome.

The intervention is performed as follows: we construct a loss function that does not promote confidence: a zero-margin hinge loss ("0-Hinge"), and optimize the network using *full-batch* gradient descent with momentum and *very small learning rate*. For an input $x$ with label $i$ the 0-Hinge loss is given by $l(x) = \sum_{j \neq i} \max(0, z_i - z_j)$ where $z_j$ represents the logit for class $j$. The form of this intervention is justified in Appendix A.7. As a control, we additionally train a model in the standard fashion using the cross-entropy loss and SGD with momentum and large initial learning rate. Since full-batch gradients are computationally expensive, we train on a subset of CIFAR10 (see Appendix A.7, where we also give the hyperparameters and learning curves.). The output margin obtained with the intervention is 5 orders of magnitude smaller than in the control experiment: $2.0 \times 10^{-4} \pm 2.0 \times 10^{-4}$ for the 0-Hinge loss and $1.6 \times 10^1 \pm 0.50 \times 10^1$ for cross-entropy loss. Figure 6 (right) compares the accuracies of the k-NN probes resulting from these training approaches. The 0-Hinge loss training achieves only a marginal improvement in accuracy (red) over an untrained network (purple), and the training split is accurately clustered only in the latest layers. This confirms the predicted behavior: the intervention leads to a model that exhibits both very small average output margins and very late clustering of the data. Very late clustering of the data implies high prediction depths since the k-NN probe classifications change in the latest layers for many data points.

## 4   Beyond a One-Dimensional Picture of Example Difficulty

In this section we transcend the one-dimensional picture of example difficulty by identifying different underlying reasons behind the difficulty of an example, in a way that is general to different architectures and datasets.

Figure 7 shows that the prediction depth can be different when an input occurs in the training split vs. the validation split. Thus, there are two axes of example difficulty:

1. Difficulty of making a prediction when an input is in the validation set
2. Difficulty of finding commonalities during training with other examples of the same ground truth class

**Both axes have a range from "clear" to "ambiguous".** In Section 3.1 we show that predictions made for validation points with later prediction depths are often inconsistent, with low consensus-

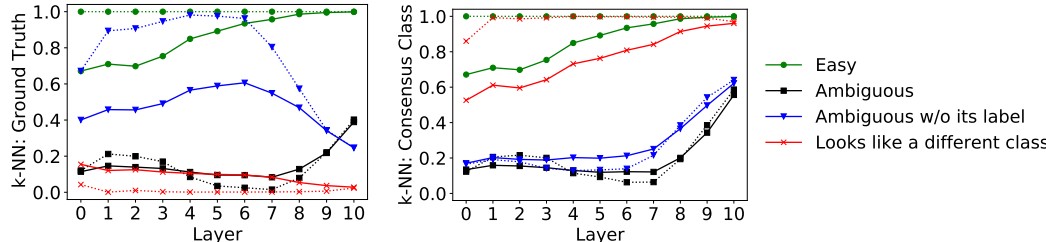

Figure 8: *Average k-NN probe confidence (solid lines) and accuracy (dotted lines) for the ground truth class (left) and consensus class (right), in the validation split for examples exhibiting extreme forms of difficulty.* Mean values for 100 examples with each form of difficulty, identified as the 100 examples closest to the corners in Figure 7 (left). This result is for CIFAR10 with ResNet18: similar plots for all datasets and architectures are shown in Appendix C.7. See Section 4 for the discussion of the result and how it can be used to improve prediction accuracy.

consistency. Conversely, a low prediction depth typically indicates an input with high consensus-consistency. For Axis 1 we will identify validation points with low prediction depths as "clear" and those with high prediction depths as "ambiguous". We will additionally identify a low or high prediction depth in the training split with examples that are respectively "clear" and "ambiguous" on Axis 2. By making combinations of low/high values of $(\mathrm{PD}_{\mathrm{Val.}}, \mathrm{PD}_{\mathrm{Train}})$ we obtain four extremes of example difficulty:

**Easy examples:** (Low $\mathrm{PD}_{\mathrm{Val.}}$, Low $\mathrm{PD}_{\mathrm{Train}}$). Such examples are often visually typical members of their class and the predicted label nearly always matches the ground truth.

**Looks like a different class:** (Low $\mathrm{PD}_{\mathrm{Val.}}$, High $\mathrm{PD}_{\mathrm{Train}}$). In the validation set, there is a clear (and nearly always incorrect) classification for such an input, but it is difficult to connect such inputs to other examples of their ground truth class during training. Mislabeled examples are of this kind, as are visually confusing images which at first appear to show something else.

**Ambiguous unless the label is given:** (High $\mathrm{PD}_{\mathrm{Val.}}$, Low $\mathrm{PD}_{\mathrm{Train}}$). These examples are difficult to connect to their predicted class in the validation split but easy to connect to their ground truth class during training. These points may, for example, visually resemble both their own class and another class. They are likely to be misclassified.

**Ambiguous:** (High $\mathrm{PD}_{\mathrm{Val.}}$, High $\mathrm{PD}_{\mathrm{Train}}$). These examples may be corrupted or show an example of a rare sub-class. Predictions for these inputs can depend strongly on the random seed used for training and initialization.

In Figure 7 we visualize CIFAR10 "Bird" images with the extreme forms of example difficulty for ResNet18, as identified using the prediction depth in the training and validation splits. In the full dataset (left panel) we see that the prediction depth can be very different in the training and validation splits: the two prediction depths are typically similar for points where the consensus class is equal to the ground truth (right panel), but can be very different when the consensus class is different from the ground truth (middle panel). This behavior is consistently reproduced for all datasets and architectures in Appendix C.6.

Looking at these examples of the class "Bird" with different difficulty types, we observe that ResNet18 finds small garden birds easiest, while birds in flight against a blue background "look like airplanes", ostriches are "ambiguous without their label" and the "ambiguous" examples are either unclear photographs or examples of rare sub-groups that don't appear frequently in the data. We found the consensus-consistency of inputs that are "Ambiguous" or "Ambiguous without its label" to be significantly lower than those of examples that are "Easy" or "Look like a different class".

In order to better understand how networks process examples with different, extreme forms of example difficulty, Fig. 8 examines how the k-NN confidence (fraction of votes) and accuracy of the ground truth class and of the consensus class progress, as validation points pass through the network. "Easy" examples are classified as their consensus class (which is equal to their ground truth class) in all k-NN probes and the confidence in the consensus class steadily increases as data points proceed through the hidden layers. Examples that "look like a different class" are also processed as members of their consensus class, similarly to "easy" examples. However, unlike "easy" examples, their consensus classes do not match their ground truth classes. Examples that are "ambiguous without their labels" are initially processed as members of their ground truth classes with intermediate confidence, but in later layers become mistaken for their consensus class. "Ambiguous" examples are

processed with low confidence and accuracy in the early layers, for both ground truth and consensus classes. In later layers "ambiguous" examples are recognized, with intermediate confidence and accuracy, as members of the consensus class, which matches the ground truth class for a sizeable fraction of "ambiguous" examples.

**Improving the prediction accuracy**    Can the prediction accuracy be improved using our understanding of how each class of difficult examples are processed by deep models? Figure 8 suggest that k-NN probes in intermediate layers may be more accurate than the full deep model for examples that are "ambiguous without their label" (data points closest to the lower right corner of Figure 7). In order to test this hypothesis, we compare the accuracy of the k-NN probe in layer 4 to the full model's prediction for the 100 examples closest to the lower right corner of Figure 8. We obtain a striking improvement in accuracy from 25% to 98% for these examples. This showcases how insights from this study can be directly used to improve prediction accuracy.

## 5   Discussion

**Summary**    We have introduced a notion of example difficulty called the prediction depth, which uses the processing of data inside the network to score the difficulty of an example. We have shown how the prediction depth is related to the accuracy and uncertainty of a prediction, the adversarial input margin and the output margin of the learned solution, and that data points that are easier according to the prediction depth are also typically learned earlier in training. We have also shown that the difficulty of an example can be both similar, or very different depending on whether an input appears in the validation split or the training split, and described four extremes of example difficulty. For data points that are "ambiguous without their label", we have demonstrated how returning the k-NN prediction in a middle layer can lead to impressive increases in model accuracy: for CIFAR10 in ResNet18 we obtained an increase in accuracy from 25% to 98% for the inputs that are most "ambiguous without their label".

**Connecting known phenomena**    In the literature, the following phenomena are separately reported from different experimental paradigms:

1. Early layers generalize while later layers memorize (Stephenson et al., 2021).
2. Model layers converge from input layer towards output layer (Raghu et al., 2017; Morcos et al., 2018).
3. Deep models learn easy data (Jiang et al., 2021; Toneva et al., 2019) and simple functions first (Hu et al., 2020; Kalimeris et al., 2019).

Following this paper, a coherent and closely related picture emerges:

1. Predictions made in early layers are more likely to be consistent than those made in later layers. Consistent predictions are likely to be correct and the expected accuracy of inconsistent predictions is naturally low (Section 3.1).
2. Data points learned early in training typically have smaller prediction depths than those learned later during training (Section 3.2).
3. On average, deep neural networks exhibit wider input and output margins (common measures of "local simplicity") in the vicinity of data with smaller prediction depths (Section 3.3).

**Pertinence of example difficulty to topics in machine learning**    Curriculum Learning attempts to treat hard examples differently from easy examples during training. Robustness to distribution shifts that change the relative frequencies of common and rare subgroups in the test set (which we have shown can have different forms of example difficulty) is important for ML Fairness. Methods developed to address heteroscedastic uncertainty typically address example difficulty as a one-dimensional quantity. We expand upon the relevance of our work to these three topics in Appendix D.

**Limitations**    We believe that the results we report stem from a deep model's representation, which is hierarchical by construction. We expect that the same results will therefore apply in larger models, larger datasets, and tasks other than image classification, but testing this remains as further work. Although we demonstrate that returning the results of a hidden k-NN can yield dramatic increases in accuracy for examples that are "ambiguous without their label", we otherwise do not explore ways to practically apply the insights we present. In particular, we expressly do not claim that all that is required for good accuracy is to reduce the prediction depth: freezing later layers of the network would not be expected to result in good generalization.

## Funding Transparency Statement

This research was funded by, and undertaken at, Google. All calculations were performed using Google's computer infrastructure.

## Acknowledgment

We would like to thank Hanie Sedghi, Ilya Tolstikhin, Ibrahim Alabdulmohsin, Daniel Keysers and Julian Eisenschlos for valuable discussions on the topic and Arthur Baldock for proofreading the manuscript.

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
