# A   Detailed Description of Experiments, Architectures and Hyperparameter Optimization

For each combination of dataset (CIFAR10, CIFAR100, Fashion MNIST, SVHN) and architecture (ResNet18, VGG16, MLP) we train 250 models with a 10% validation split selected at random each time, and an additional 25 models on the full training set.

## A.1   Datasets

CIFAR10 / CIFAR100:
Reference: (Krizhevsky et al., 2009). License: MIT.
URL: `https://www.cs.toronto.edu/~kriz/cifar.html`

Fashion MNIST:
Reference: (Xiao et al., 2017). License: MIT.
URL: `https://github.com/zalandoresearch/fashion-mnist`

Street View House Numbers (Cropped Digits):
Reference: (Netzer et al., 2011). License: CC0.
URLs: `http://ufldl.stanford.edu/housenumbers/`
`https://www.kaggle.com/stanfordu/street-view-house-numbers`

## A.2   Architectures

### A.2.1   ResNet18

We implemented the standard ResNet18 architecture for CIFAR10 (He et al., 2016), except that we replaced Batch Norm with Group Norm and applied Weight Standardization, following recent state of the art (Kolesnikov et al., 2020).

### A.2.2   VGG16

We used VGG16 (Simonyan and Zisserman, 2015), except that we removed the final three dense layers: a standard modification for datasets smaller than ImageNet. We also did not use batch-norm or dropout: our focus is on understanding trends in example difficulty and we do not expect the results to be dependent on these devices.

### A.2.3   MLP

Our MLP architecture comprises seven hidden layers with ReLU activations. We chose seven layers after performing the experiments shown in Figure A.9. There we show the accuracies of k-NN probes placed after each operation of two MLP architectures, depths 15 layers and 7 layers, both of width 2048. We used CIFAR10 with 40% fixed random label noise as a reasonably difficult model classification task, to choose the depth.

### A.2.4   Data augmentation

We did not apply data augmentation: different data augmentation schemes could be expected to have disparate effects on different examples, but we do not expect them to change the overall phenomena that we report here. We leave the use of data augmentation to subsequent studies.

## A.3   Hyperparameter optimization

For each architecture and dataset we initially performed $10^4$ steps of SGD with momentum, using all combinations of the following hyperparameters: learning rate $\in [4 \times 10^1, 1 \times 10^{-1}, 4 \times 10^{-2}, 1 \times 10^{-2}, 4 \times 10^{-3}, 1 \times 10^{-3}, 4 \times 10^{-4}, 1 \times 10^{-4}]$ ; momentum $\in [0.0, 0.5, 0.9, 0.95]$; weight decay $\in [0, 5 \times 10^{-4}]$. In CIFAR10, we additionally considered a learning rate of $2 \times 10^{-2}$. For each dataset and architecture we selected the 7 most accurate and stable training curves, extended the number of training steps and added a learning rate schedule, reducing the learning rate in steps of $\frac{1}{5}$. At least two rounds of optimization were performed to adapt the learning rate schedule for each combination of architecture and dataset. In each case a mini-batch size of 256 was used. The final parameters obtained are shown in Table 1, which also gives the hyperparameters used in Sec. 3.3 and Appendix A.2.3 for CIFAR10 with 40% label noise. Final accuracies of the trained models are given in Table 2.

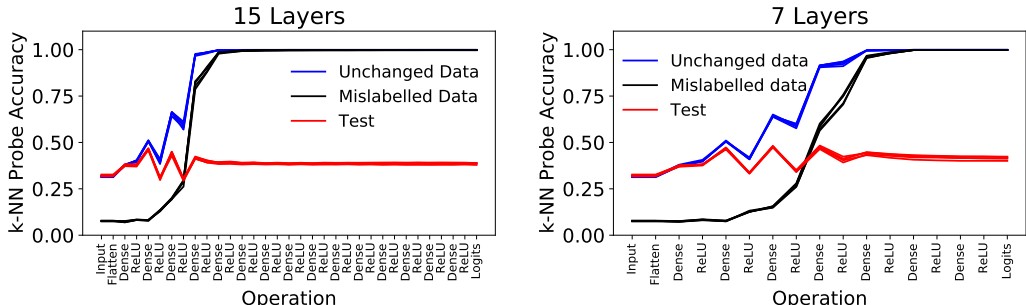

Figure A.9: *Seven layers are sufficient in MLP for CIFAR10 with 40% random label noise.* CIFAR10 with 40% random label noise. For this plot, k-NN probes were placed after every operation in two MLP architectures of the same width (2048) but different depths. **Left:** 15 dense layers; **Right:** 7 dense layers. Separate accuracies are reported for the test split, those data points in the training split with unchanged labels and the randomly mislabeled data in the training split.

| | Learning Rate | Momentum | Weight Decay | Schedule / steps |
|---|---|---|---|---|
| | SVHN | | | |
| ResNet18 | $4 \times 10^{-2}$ | 0.95 | 0.0 | [7000] |
| VGG16 | $4 \times 10^{-2}$ | 0.9 | 0.0 | [3000, 6000, 1000] |
| MLP | $4 \times 10^{-2}$ | 0.9 | 0.0 | [2500, 5500, 2000] |
| | Fashion MNIST | | | |
| ResNet18 | $1 \times 10^{-2}$ | 0.95 | 0.0 | [4000, 3000] |
| VGG16 | $1 \times 10^{-2}$ | 0.95 | 0.0 | [3000, 6000, 1000] |
| MLP | $4 \times 10^{-2}$ | 0.5 | 0.0 | [10000, 2500] |
| | CIFAR10 | | | |
| ResNet18 | $4 \times 10^{-2}$ | 0.95 | 0.0 | [7000] |
| VGG16 | $4 \times 10^{-2}$ | 0.9 | 0.0 | [5000, 1000] |
| MLP | $2 \times 10^{-2}$ | 0.9 | 0.0 | [5000, 1250, 1000] |
| | CIFAR10 w/ 40% (Fixed) Randomized Labels | | | |
| VGG16 | $4 \times 10^{-2}$ | 0.9 | 0.0 | [5000, 10000] |
| MLP | $2 \times 10^{-2}$ | 0.9 | 0.0 | [12000, 1250, 4000 |
| | CIFAR100 | | | |
| ResNet18 | $1 \times 10^{-1}$ | 0.95 | 0.0 | [6000] |
| VGG16 | $4 \times 10^{-2}$ | 0.9 | 0.0 | [2500, 7500] |
| MLP | $1 \times 10^{-1}$ | 0.95 | 0.0 | [2500, 6000, 1500] |

Table 1: Training parameters for each model and dataset.

## A.4 Convergence and consistency of k-NN probe accuracies

We tested the convergence of k in k-NN for VGG16 on CIFAR10. Figure A.10 shows the accuracies of k-NN probes after every operation of the network for $k \in [3, 10, 30]$. We see that these k-NN probe accuracies are insensitive to $k$ for $k = 30$.

Figure A.9 shows separate results for five independent training runs. Similarly, Figure 5 (right) and Figure 6 (right) each show the mean and uncertainty on the k-NN probe accuracies from 5 independent runs. The spread of results in these figures is tight, demonstrating consistency of the results.

## A.5 Placement of k-NN probes

For prediction depth, in MLP we constructed k-NN probes after the dense operations and the softmax, in VGG16 after the convolutions and softmax, and in ResNet18 we constructed the probes after the initial Group Norm operation, the sum operations at the end of each block and after the softmax operation.

| | SVHN | Fashion MNIST | CIFAR10 | CIFAR100 |
|---|---|---|---|---|
| ResNet18 | 95% | 93% | 83% | 56% |
| VGG16 | 95% | 93% | 83% | 45% |
| MLP | 85% | 90% | 59% | 29% |

Table 2: Final accuracies of the trained models.

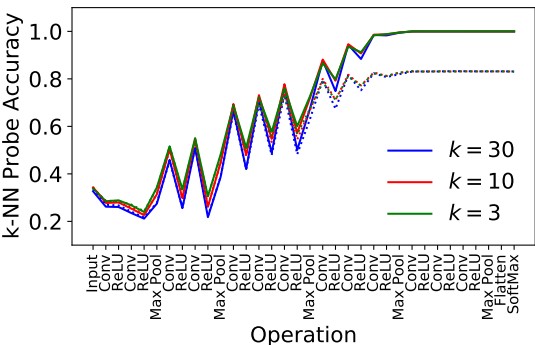

Figure A.10: CIFAR10, VGG16. k-NN probe accuracies after each operation for $k \in [3, 10, 30]$. Solid lines: training set. Dotted lines: test set. Differences in these results are comparable to the scatter observed for networks trained with different random seeds at $k = 30$.

From figures A.9 and A.10 it is clear that there are upper and lower envelopes that bound the k-NN probe accuracies: the lower envelope corresponds to the ReLU activations and the upper envelope to the operations immediately preceding them. We chose the preceding operations which, in effect, conceptually shifts the ReLU activations to the "start" of a layer rather than the "end" of the preceding layer.

### A.6 Notes on definitions

#### A.6.1 Consistency of the model's prediction with the k-NN probe after the softmax layer

Deep classifier models are trained to create linear separation of the classes in the softmax layer. There is nearly perfect agreement between the k-NN probe after the softmax layer and predictions of the full model. In the rare case that the k-NN probe after the softmax predicts a different class from the full network we do not assign a prediction depth. Such data points are extremely rare: we found zero such data points in the large majority of models and always fewer than 1 in $10^4$.

#### A.6.2 Tiebreaks in the consensus class

When obtaining the consensus class, if predictions are tied between more than one class and the ground truth is in the tiebreak, then we break the tie in favor of the ground truth class. If the ground truth is not in the tiebreak then we report the tied class with the lowest integer index. This choice was motivated by ease of implementation. We are confident that the overall results we report are unaffected by this choice.

#### A.6.3 Estimating the consensus-consistency

We used the same ensemble to obtain both the consensus class $\hat{y}_A(x)$ and the consensus-consistency score. Thus we are reporting relationships between observables for a given ensemble. This is a biased estimator of (2): an unbiased estimator could have been constructed by training an additional set of models to obtain the consensus class, but at greater cost. We are confident that this does not affect the conclusions of this study.

### A.7 Justification and hyperparameters for the output margin intervention

A number of published works informed the design of our intervention. Firstly, Soudry et al. (2018) demonstrate that the cross-entropy (CE) loss leads to large margins. In contrast to the cross-entropy, the 0-Hinge loss has zero gradient if the prediction is correct, so it does not push the model to

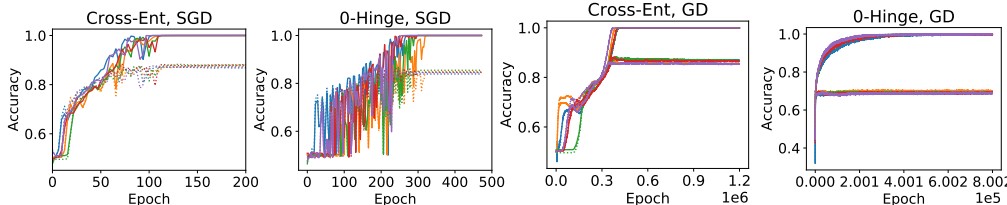

Figure A.11: *Training curves for Cross-Entropy and 0-Hinge Losses, with either SGD with momentum and large initial learning rate, or GD with momentum and a small learning rate.* The initial learning rates and schedules are set to obtain nearly smooth learning curves for GD and noisy learning curves for SGD. Each plot shows five separate learning curves. Solid lines show training accuracies and dotted lines show test accuracies.

become arbitrarily confident. Secondly, Keskar et al. (2017) show that smaller batch sizes lead to the discovery of flatter minima, which also corresponds to a wider margin (Neyshabur et al., 2017). Thirdly, Keskar et al. (2017), Smith and Le (2018) and Smith et al. (2018) show that the gradient noise level in stochastic gradient descent is proportional to $\frac{\text{Learning Rate}}{\text{Batch Size}}$. Having an appreciable noise level early in training plays an important role in finding the flatter minima with larger output margins reported in Keskar et al. (2017). Our intervention to minimize the margin therefore combines both of the following changes:

1. Changing the loss from cross-entropy to the 0-Hinge loss
2. Minimizing the learning rate and making the batch size as large as possible

To test whether both or only one of these changes is required to obtain small output margins, we performed separate runs, without any intervention, applying the changes individually and applying them together. The starting point (the control) is training with cross-entropy loss and SGD with momentum and large initial learning rate.

We trained VGG16 on CIFAR10. The hyperparameters, presented in Table 3, were set for each loss, to obtain nearly smooth learning curves for full-batch gradient descent and very noisy learning curves for SGD. In Figure A.11 we show the learning curves for these models. Since full-batch gradients are expensive to compute we restricted the experiments to separating two classes ("Horse" and "Deer") with 4096 training images in total (evenly split).

| Name | Batch Size | Initial Learning Rate | Schedule / Steps | Momentum |
|---|---|---|---|---|
| CE, SGD | 256 | $4 \times 10^{-3}$ | $[3200]$ | 0.9 |
| CE, GD | 4096 | $6.4 \times 10^{-6}$ | $[1.2 \times 10^6]$ | 0.9 |
| 0-Hinge, SGD | 256 | $4 \times 10^{-2}$ | $[5000, 2500]$ | 0.9 |
| 0-Hinge, GD | 4096 | $6.4 \times 10^{-5}$ | $[8 \times 10^5]$ | 0.95 |

Table 3: Hyperparameters for all combinations of CE vs. 0-Hinge loss and SGD with momentum and large initial learning rate vs. GD with momentum and small learning rate. In the learning rate schedules we reduced the learning rate by a factor of $\frac{1}{5}$ for each new set of training steps. Weight decay was not employed in these calculations since we do not expect typical, modest amounts of weight decay to qualitatively affect the results.

Table 4 lists the mean accuracy and output margin for all four combinations of loss function and optimizer. We can see that the combination of both changes yields the smallest mean output margin, $10^2$ times smaller than the next smallest margin. Figure A.12 presents the k-NN probe accuracies in the hidden layers for all four combinations of loss and optimizer. The combined intervention, which has the smallest margin, leads to the data being accurately clustered in the very latest layers.

## B  Further Related Work

Previous studies of deep learning on the level of individual data points have: sought to explain its accuracy by focusing on the interference of per-example gradients during training (Chatterjee, 2019; Zielinski et al., 2020); improved our understanding of deep learning by studying its performance on datasets with partially randomized labels, which corresponds to a specific binary partitioning of example difficulty Arpit et al. (2017); quantified example difficulty using 5 different observables:

| Name | Mean Accuracy | Mean Output Margin |
|------|---------------|--------------------|
| CE, SGD | 87.6% | $1.6 \times 10^1$ |
| CE, GD | 86.7% | $1.1 \times 10^1$ |
| 0-Hinge, SGD | 83.9% | $6 \times 10^{-2}$ |
| 0-Hinge, GD | 69.5% | $2.0 \times 10^{-4}$ |

Table 4: Mean accuracy and output margin for CE vs. 0-Hinge losses and SGD with momentum and large initial learning rate vs. GD with momentum and small learning rate.

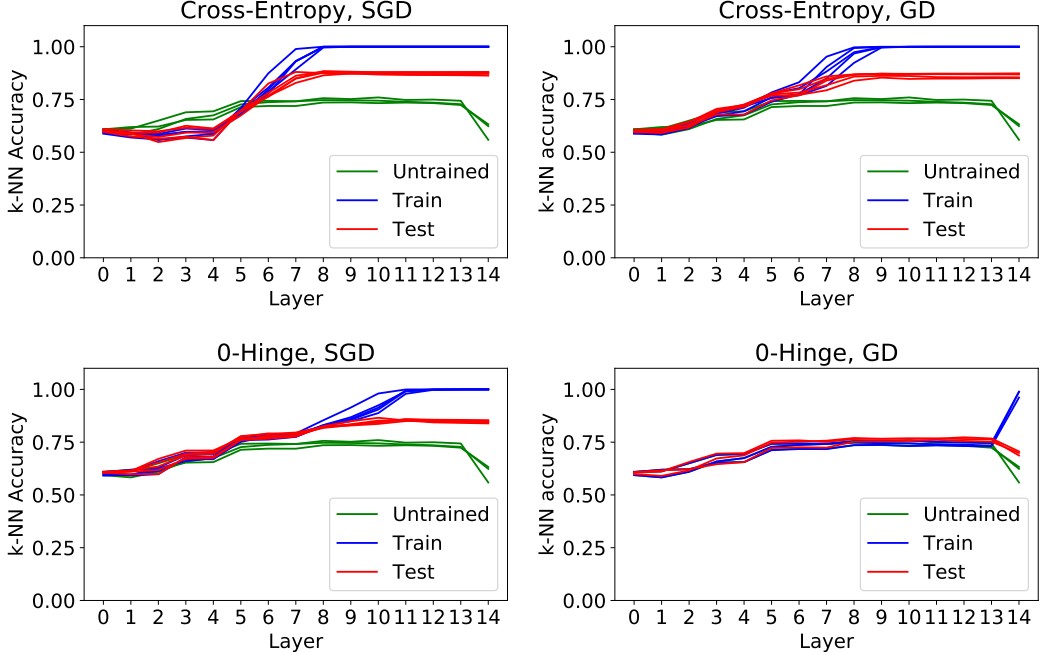

Figure A.12: *Accuracies of k-NN probes in the hidden layers of VGG16, resulting from each combination of Cross-Entropy vs. 0-Hinge loss and SGD with momentum and large initial learning rate vs. GD with momentum and small learning rate.* In each case we compare to the probes for untrained (freshly initialized) networks. Only the 0-Hinge with gradient descent using momentum and small learning rate ("0-Hinge, GD") leads to clustering in the latest layers.

1) the change in a network's output for elements of the training set after subsequent fine-tuning on a disjoint dataset, 2) the adversarial input margin of an example, 3) the agreement of models in an ensemble, 4) the average confidence of models in an ensemble, and 5) the disparate impact of differential privacy Carlini et al. (2019); identified difficult examples with those disproportionately impacted by pruning and compression Hooker et al. (2019), with those whose classifications are more often forgotten during training Toneva et al. (2019), and with those that are least likely to be correctly classified in the validation set Jiang et al. (2021); demonstrated a correspondence between those examples that a human finds difficult and examples a machine finds difficult Lalor et al. (2018). In contrast to these works, we study the computational difficulty of inferring the class of an input: the amount of computation used to connect that input with its class label inside the network. Our definition of example difficulty is precisely described in Section 2.

In Hacohen et al. (2020) the authors report that the order during training in which data points are learned is common between different architectures and random seeds in deep learning. In light of the correlation between prediction depth and the order of learning data points (as reported in Section 3.2), their result reflects the sanity checks performed in Section 2.2: that prediction depth is consistent between architectures and random seeds.

Distinct from the forms of example difficulty we describe in Section 4, Hooker et al. (2019) propose four different forms of example difficulty: "ground truth label incorrect or inadequate", "multiple-

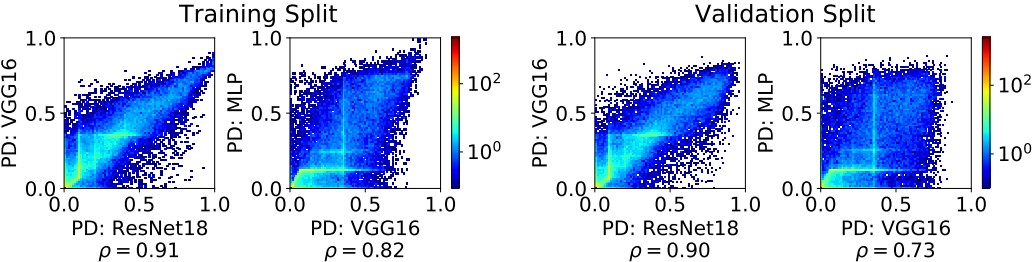

Figure C.13: *Consistency of prediction depth between architectures for SVHN.* Histograms comparing the mean value of prediction depth obtained for each data point, across the ensemble of trained models. Left pair: training split. Right pair: validation split. Spearman's Correlation Coefficient is given beneath each plot. See Appendix C.2 for details.

object image", "corrupted image", "fine-grained classification". The forms of difficulty we describe in this paper follow directly from the computational difficulty of the examples, derived from the model's behavior. In contrast, Hooker et al. (2019) employ intuitive notions of difficulty to define their four forms and ask humans to assign difficult examples to these categories.

The Deep k-Nearest Neighbors method Papernot and McDaniel (2018) builds a series of k-NN probes in the hidden spaces of the network. When a test example is processed by the network, Deep k-NN identifies the nearest neighbors of the example in every layer, and then classifies the example according to the class labels of the aggregated nearest neighbors. By comparing the number of neighbors the example has of the predicted class to the number of similarly labeled nearest neighbors that were recorded (across all layers) for examples in a hold-out test set, Deep k-NN is able to quantify the probability that the prediction is correct and to identify OOD examples. However, the authors do not report the phenomena reported here. Our results may yet enable the development of new Deep k-NN methods. Another algorithm Bahri et al. (2020) constructs a k-NN probe in the logit space of a network, and demonstrates that this enables improved detection of mislabeled data.

# C    Consistency of the Main Results Reported in the Paper

## C.1    Prediction depth corresponds to an intuitive notion of example difficulty

Table 5 presents representative examples with either extremely high or extremely low prediction depths in ResNet18. Examples are taken from two randomly chosen classes of each dataset. The images shown provide further evidence for our claim in Figure 1 that examples predicted in the input are visually typical ("easy"), while those predicted in the softmax are mislabeled and/or visually confusing ("hard" examples).

## C.2    Consistency of prediction depth between architectures

To visually reinforce the correlations reported in Figure 2 (right), Figures C.13 to C.16 reproduce the result from Figure 2 (right) for all datasets in both the training and validation splits. For each combination of dataset and architecture we trained 250 models with random 90:10% training:validation splits as described in Appendix A. These histograms compare the mean prediction depths of the data points between different architectures. Separate plots are shown for the training and validation splits. In each case we've rescaled prediction depth to the interval $[0, 1]$ for visual ease of comparison between datasets. Each histogram is accompanied by the corresponding Spearman's Correlation Coefficient.

## C.3    Relationship between prediction depth and prediction consistency

Figures C.17 and C.18 reproduce the results of Figure 3 and Figure 4 (left) for every dataset and architecture. The gradients of the linear bounds reported in the paper depend on the difficulty of the classification task: easier tasks are solved after fewer layers.

Figure C.19 reproduces Figure 4 (middle) for every dataset and architecture. Similarly, Figure C.20 reproduces Figure 4 (right) for all datasets and architectures. Related to Figure C.19, in Figure C.21 we show that the prediction depth in one model can be used to estimate the prediction entropy of

| Dataset / Class | Low PD Examples | High PD Examples |
|---|---|---|
| CIFAR100 / BICYCLE |  |  |
| CIFAR100 / TROUT |  |  |
| CIFAR10 / AIRPLANE |  |  |
| CIFAR10 / TRUCK |  |  |
| Fashion MNIST / DRESS |  |  |
| Fashion MNIST / SANDAL |  |  |
| SVHN / 5 |  |  |
| SVHN / 8 |  |  |

Table 5: This table presents additional examples to support the claim made in Figure 1, that examples predicted in the input are visually typical ("easy"), while those predicted in the softmax are mislabeled and/or visually confusing ("hard" examples). We report training inputs for two random classes from each of the CIFAR100, CIFAR10, Fashion MNIST, and SVHN datasets. The middle column shows training examples with very low prediction depths in ResNet18. The right-hand column shows training examples with very high prediction depths in ResNet18.

an ensemble of models, where members of the ensemble have the same architecture and are trained using the same hyperparameters but with different random seeds.

**Prediction entropy:** The entropy of predictions in an ensemble for an unseen input $x$. Consider an ensemble of models trained on $r$ random subsets of the complete dataset $\tilde{\mathcal{S}} \sim \mathcal{S} \backslash \{(x, y)\}$ (which explicitly do not include $(x, y)$). We obtain the normalized histogram of the one-hot predictions of this ensemble for the input $x$. The prediction entropy is the entropy of that histogram. For $N$ classes the entropy of the prediction histogram is given by

$$S(x) = -\sum_{i=1}^{N} p_i(x) \log p_i(x) \qquad (3)$$

where $p_i(x)$ represents the fraction of models that predicted the class $i$ for input $x$.

Figure C.22 shows the histogram of average prediction depth (validation set) vs. prediction entropy for each dataset and architecture. We remark that the mean prediction depth defines a linear upper bound on the prediction entropy similar to the corresponding linear lower bound on the consensus-consistency score (Figures C.17 and C.18).

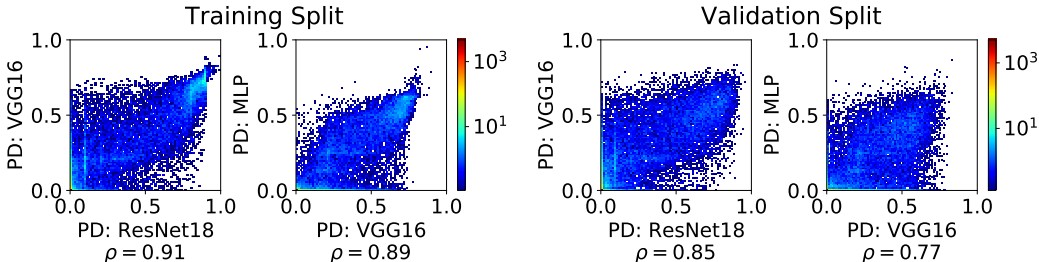

Figure C.14: *Consistency of prediction depth between architectures for Fashion MNIST.* Histograms comparing the mean value of prediction depth obtained for each data point, across the ensemble of trained models. Left pair: training split. Right pair: validation split. Spearman's Correlation Coefficient is given beneath each plot. See Appendix C.2 for details.

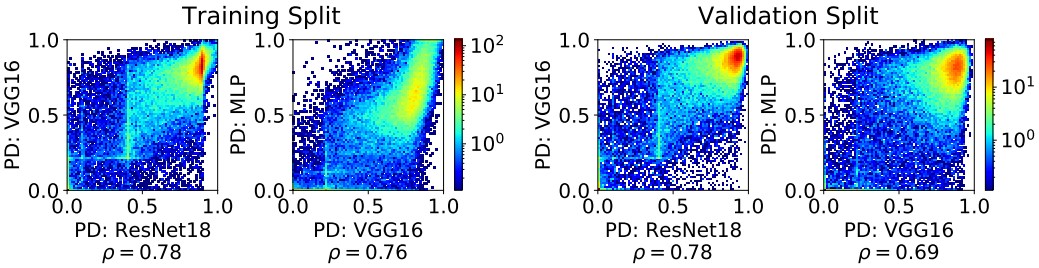

Figure C.15: *Consistency of prediction depth between architectures for CIFAR100.* Histograms comparing the mean value of prediction depth obtained for each data point, across the ensemble of trained models. Left pair: training split. Right pair: validation split. Spearman's Correlation Coefficient is given beneath each plot. See Appendix C.2 for details.

### C.4 Comparison of prediction depth and iteration learned

Figure C.23 reproduces the result shown in Figure 5 (left) for every architecture and dataset. To give a more complete picture of the relationship between the prediction depth and the iteration learned, Figures C.24 to C.27 show histograms of the mean prediction depth and iteration learned for each data point when it occurs in both the training and validation splits. As described in Appendix A, for each dataset and architecture we trained 250 models with random 90:10% validation:train splits. Each time a data point appears in either split we record the prediction depth and the iteration learned. These histograms compare the mean prediction depth to the mean iteration learned for all data points in both the train and validation splits. The Spearman's Correlation Coefficient is given beneath each plot.

### C.5 Consistency of margin results

Figures C.28 to C.31 reproduce Figure 6 (left and middle) for all datasets and architectures in both the training and test splits.

### C.6 Consistent two-dimensional relationship between prediction depths in the training and validation splits

Figures C.32 to C.35 demonstrate consistency of the histograms shown in Figure 7 for all datasets and architectures. As described in Appendix A, for each dataset and architecture we trained 250 models with random 90:10% validation:train splits. Each time a data point appears in either split we record the prediction depth. These histograms compare the mean prediction depths in the two splits for all data points which can be very different from each other, depending on whether the consensus class matches or differs from the ground truth class.

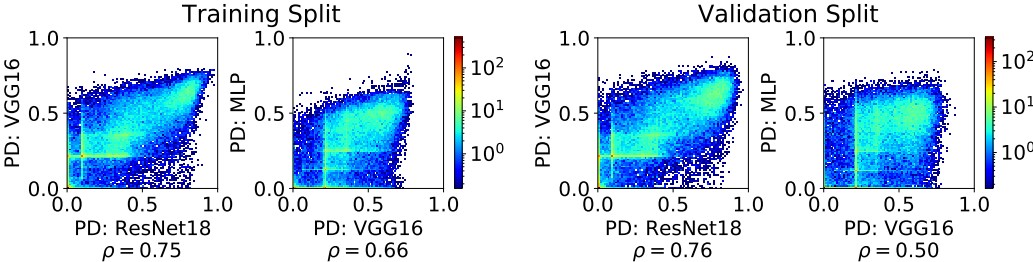

Figure C.16: *Consistency of prediction depth between architectures for CIFAR10.* Histograms comparing the mean value of prediction depth obtained for each data point, across the ensemble of trained models. Left pair: training split. Right pair: validation split. Spearman's Correlation Coefficient is given beneath each plot. See Appendix C.2 for details.

### C.7 Evolution of clustering in the hidden layers for the different forms of example difficulty

Figures C.36 to C.47 reproduce similar behavior to that shown in Figure 8 for all datasets and architectures. Please see Figure 8 for a detailed description.

## D Pertinence of example difficulty to topics in machine learning

We will describe the relevance of our work to distribution shift and robustness; algorithmic fairness, curriculum learning and models that explicitly address heteroscedastic uncertainty.

**Distribution Shift and Robustness:** Recent work has hypothesized that the linear relationship between the performance of a model before and after distribution shift could potentially be explained in a theory based on the difficulty of examples (Recht et al., 2019). Recent work has additionally discussed how examples that belong to a minority group might appear difficult to classify correctly under distribution shift (Nagarajan et al., 2021). Therefore it seems natural to suppose that the richer picture of example difficulty we introduce could lead to a deeper understanding of distribution shift and aid with the development of more robust algorithms.

**Curriculum Learning:** This class of training algorithms exploits additional information about a dataset (obtained in advance) to present easier examples earlier in the training process (El-man, 1993; Sanger, 1994; Bengio et al., 2009). Different notions of difficulty have been the subject of several related studies (Bengio et al., 2009; Toneva et al., 2019; Hacohen and Weinshall, 2019) and it has been shown that (neglecting the cost of obtaining the curriculum) following a curriculum can improve training time significantly, particularly for large training data (Wu et al., 2021). We envisage that richer, more effective curricula could be designed by distinguishing different forms of example difficulty. This could, for example, be achieved setting the curriculum according to a each data point's location in Figure 7.

**Algorithmic Fairness:** We have seen that mislabeled data is processed similarly to data that simply looks mislabeled to the algorithm (both "look like a different class"). This presents a fairness challenge when filtering "noisy labels". Similarly, we have seen that examples of rare subgroups (which are essential to include in the training set for robustness (Feldman and Zhang, 2020) and fairness (Hooker et al., 2020) are processed similarly to truly "ambiguous" inputs. Finding ways to deal with "label noise" without biasing against these subgroups remains an open challenge. In further work, we anticipate that examining datasets in an enlarged space of different example difficulty measures (Jiang et al., 2021; Toneva et al., 2019; Carlini et al., 2019; Hooker et al., 2019; Lalor et al., 2018; Agarwal and Hooker, 2020) may allow algorithms that distinguish between these different sources of label noise to reach higher accuracy and to be fairer.

**Heteroscedastic Uncertainty:** There are a class of models with two heads, one to model the mean and the other the uncertainty of the prediction (E.g. Kendall and Gal (2017); Kendall et al. (2018)). These models learn to become uncertain on difficult inputs and treat example difficulty as a one-dimensional quantity. It seems highly likely that this uncertainty will lead to the model down-weighting examples of rare subgroups in the data. We suggest

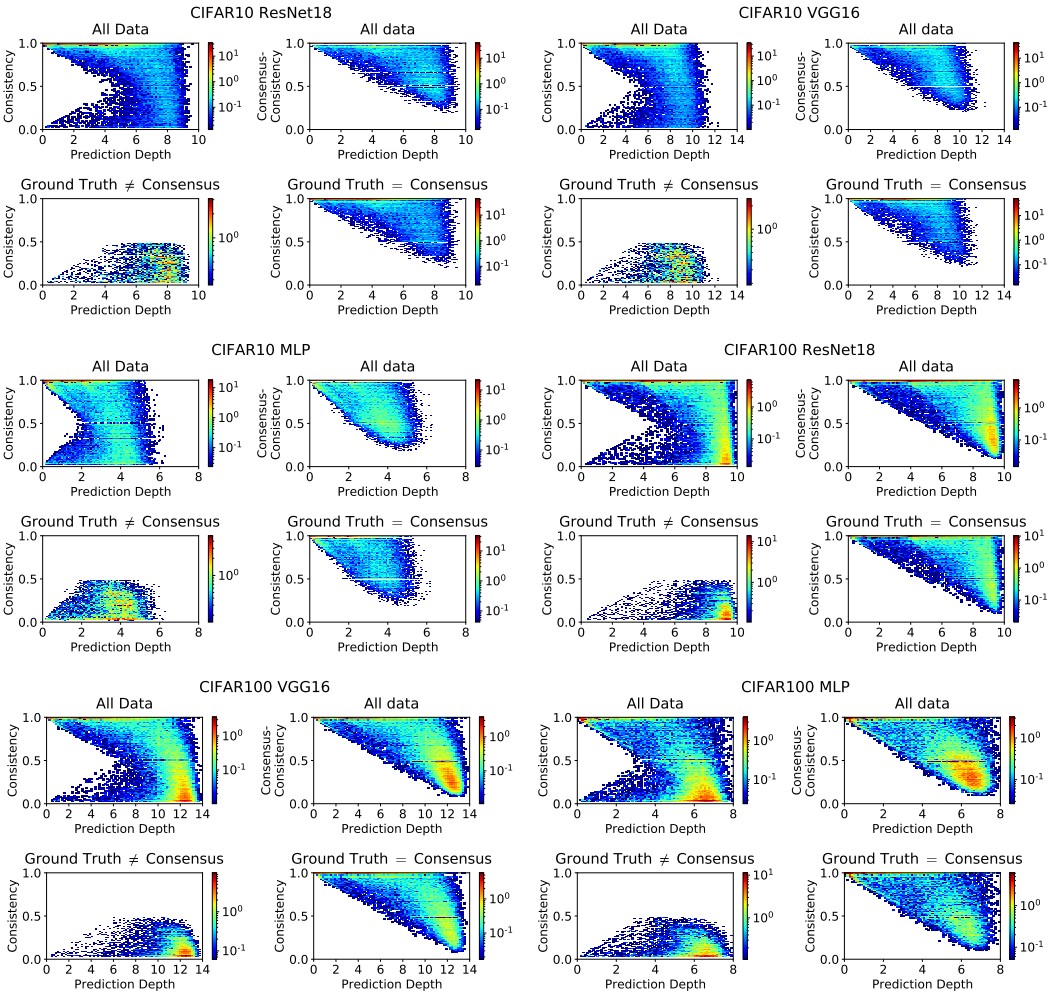

Figure C.17: This figure demonstrates the consistency of the behavior shown in Figure 3 and Figure 4 (left) for all architectures with CIFAR10 and CIFAR100.

that methods for modeling uncertainty could additionally be tasked with estimating the location of a training point in Figure 7. It seems plausible to suppose that new models able to distinguish the form of an example's difficulty could later be refined to be fairer, more accurate and better calibrated.

# E    Alternative Definitions for Prediction Depth

Instead of using the network's final prediction on a data point to assign the prediction depth, one could instead use the ground truth label. This would require a different rule for assigning a prediction depth to validation data points that are incorrectly classified as compared to data points that are correctly classified. We consider our definition to be simpler than combining two separate rules.

One could alternatively have defined the prediction depth for each example by first leaving it out of the training set, and then training networks of different depths to identify the number of layers required to classify it correctly. In fact, architectures of different depths have different inductive biases, so the relative difficulty of inputs can become inverted with changing depth (Mangalam and Prabhu, 2019). Such an approach would be expensive but could lead to a rich picture of how example difficulty changes with architecture.

Another potential approach would have been to use a linear classifier such as Logistic Regression in the embedding spaces. Indeed linear probes, logistic regression and SVM probes have been previously applied to the hidden spaces of DNNs (E.g. Cohen et al. (2018); Alain and Bengio (2017)).

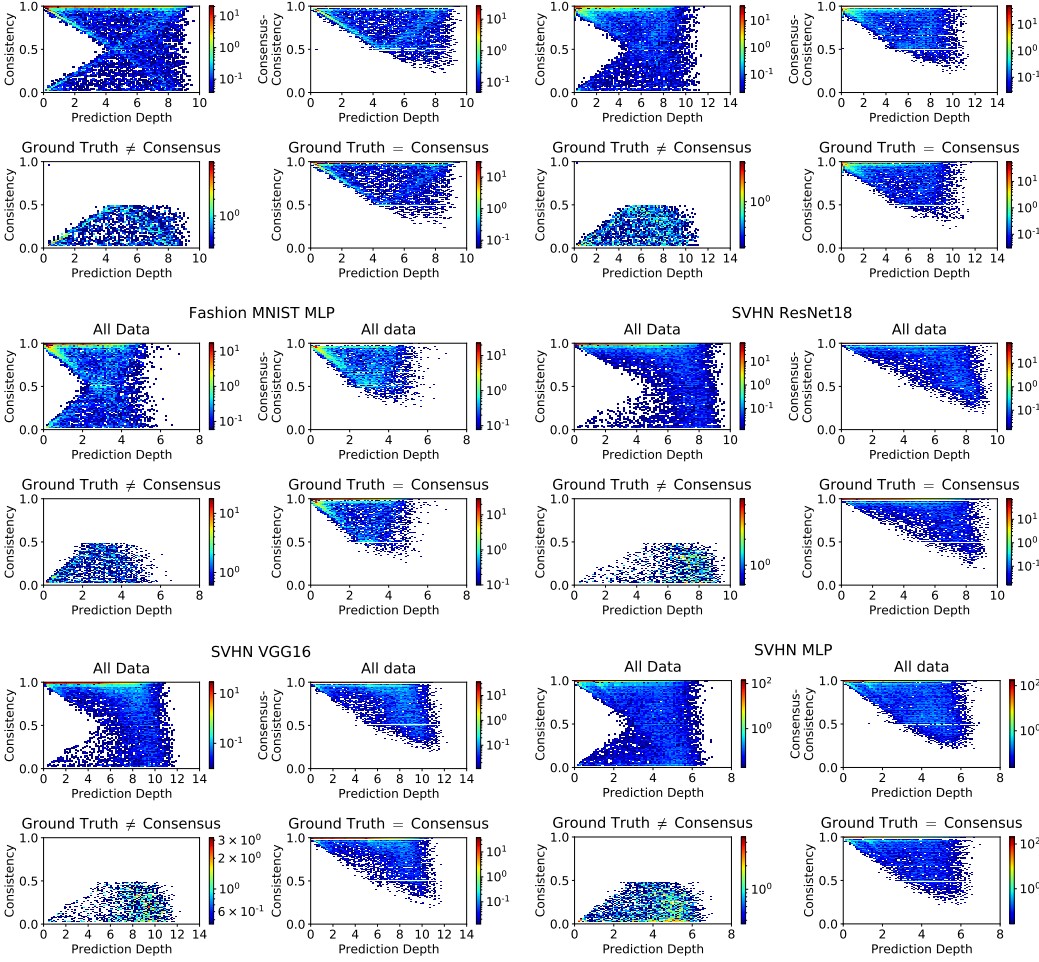

Figure C.18: This figure demonstrates the consistency of the behavior shown in Figure 3 and Figure 4 (left) for all architectures with Fashion MNIST and SVHN.

On advantage of linear, logistic regression and SVM probes, is that they have a fixed inference cost independent of the size of the training set. This is not the case for k-NN in high dimensional embedding spaces. For a sufficiently large training split, one could select a fixed random subset of the training split to use as the support set for the k-NN probes. This would reduce the cost of evaluating the k-NN probes. Doing so would affect the value of the prediction depth in individual data points but should not affect the broad conclusions of this paper.

Figure E.48 compares the behavior of k-NN probes and Logistic Regression (LR) probes after the convolution operations of VGG16 with CIFAR10. LR is able to completely separate the training set after the first convolution operation. We also show the behavior when training LR on a random 50% of the dataset and predicting on the other half. k-NN shows lower accuracy until the classes become entirely clustered. We chose k-NN probes for this investigation.

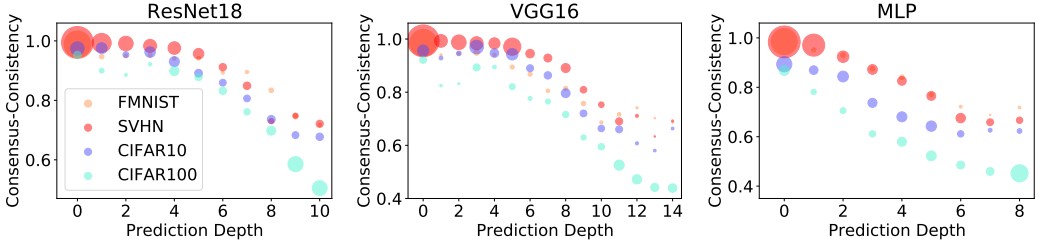

Figure C.19: This figure demonstrates the consistency of the result shown in Figure 4 (middle) for all datasets and architectures.

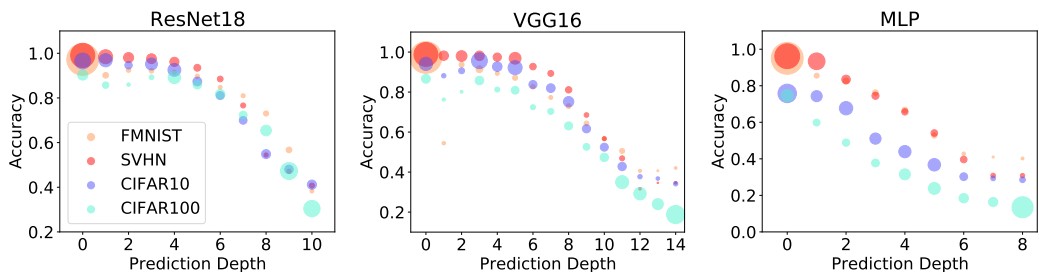

Figure C.20: This figure demonstrates the consistency of the result shown in Figure 4 (right) for all datasets and architectures.

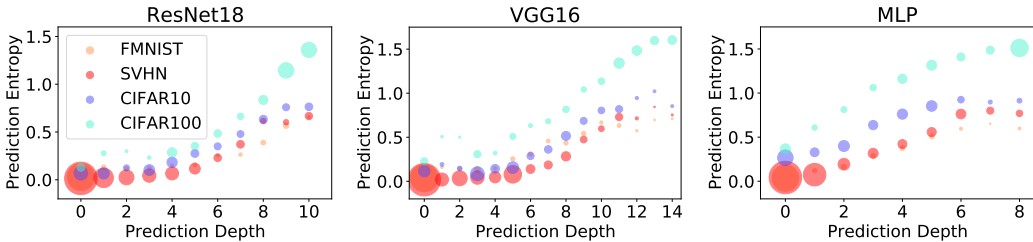

Figure C.21: *The prediction depth in one model can be used to estimate the prediction entropy of an ensemble.* The size of the marker indicates the fraction of data points with each prediction depth. We trained 25 models on each dataset and architecture with different random seeds. We take the prediction depth from one trained model and report the average prediction entropy of the corresponding data points, where the prediction entropy is determined from the remaining 24 models. As in Figure C.19, predictions for data points with smaller prediction depths have lower mean entropy (are more consistent) than those of data points with larger prediction depths.

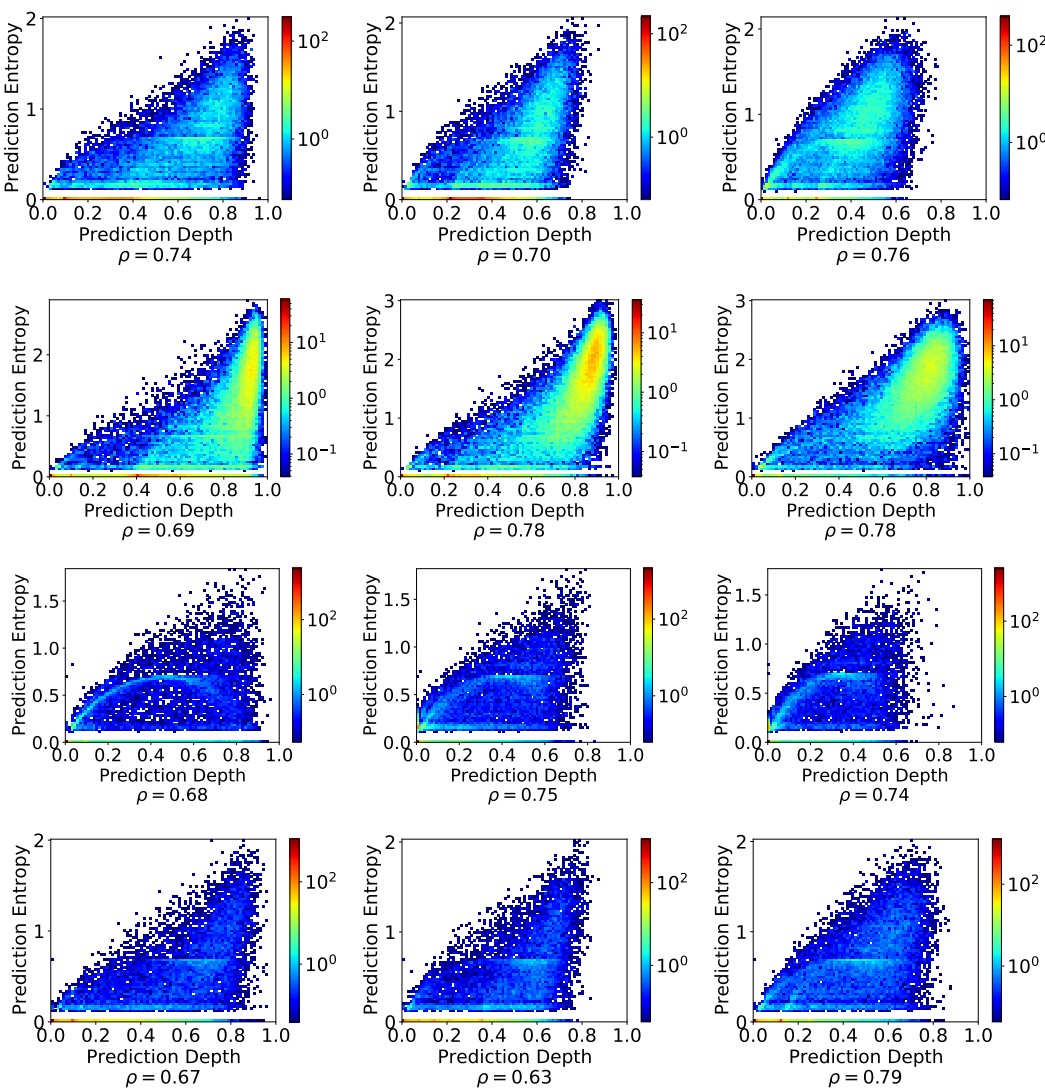

Figure C.22: First (Top) Row: CIFAR10. Second Row: CIFAR100. Third Row: Fashion MNIST. Fourth (Bottom) Row: SVHN. Left Column: ResNet18. Middle Column: VGG16. Right Column: MLP. Histograms showing consistency of the relationship between prediction depth in the validation set and prediction entropy of an ensemble. As described in Appendix A, for each dataset and architecture we trained 250 models with random 90:10% validation:train splits. Each time a data point appears in the validation split we record the prediction depth and the prediction. These histograms compare the average prediction depth for each data point to its prediction entropy. We observe that the prediction depth gives linear upper bounds for the prediction entropy as it does linear lower bounds for the consensus-consistency (Figures C.17 and C.18).

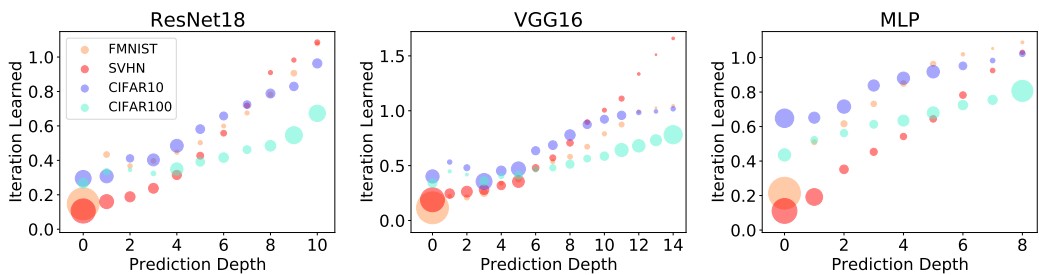

Figure C.23: This figure demonstrates the consistency of the result shown in Figure 5 (left) for all datasets and architectures.

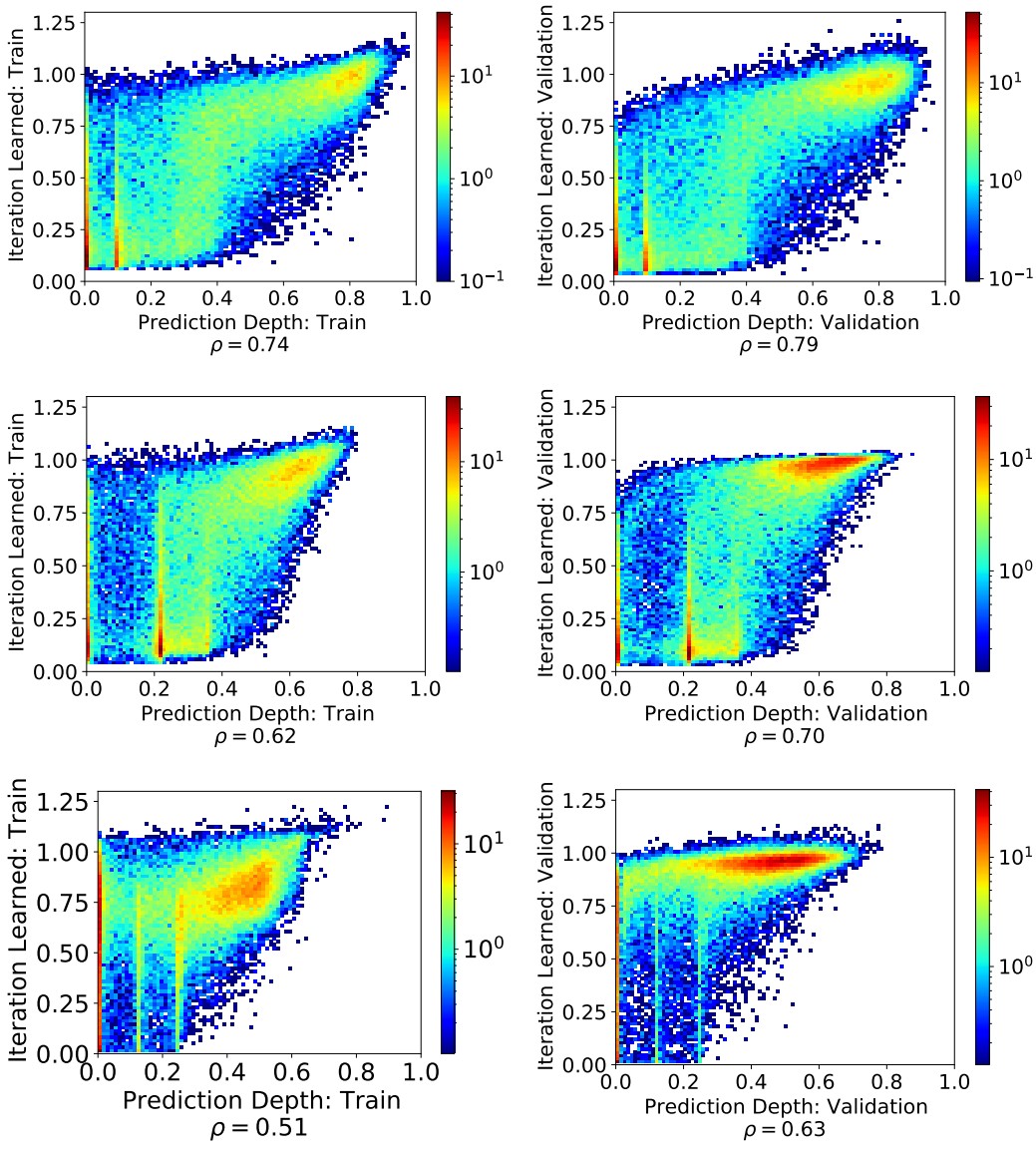

Figure C.24: CIFAR10. Top row: ResNet18. Middle row: VGG16. Bottom row: MLP. Histogram comparing the mean prediction depth to the mean iteration learned when each data point occurs in either the training split (left column) or the validation split (right column). See Appendix C.4 for a description of the experiments performed.

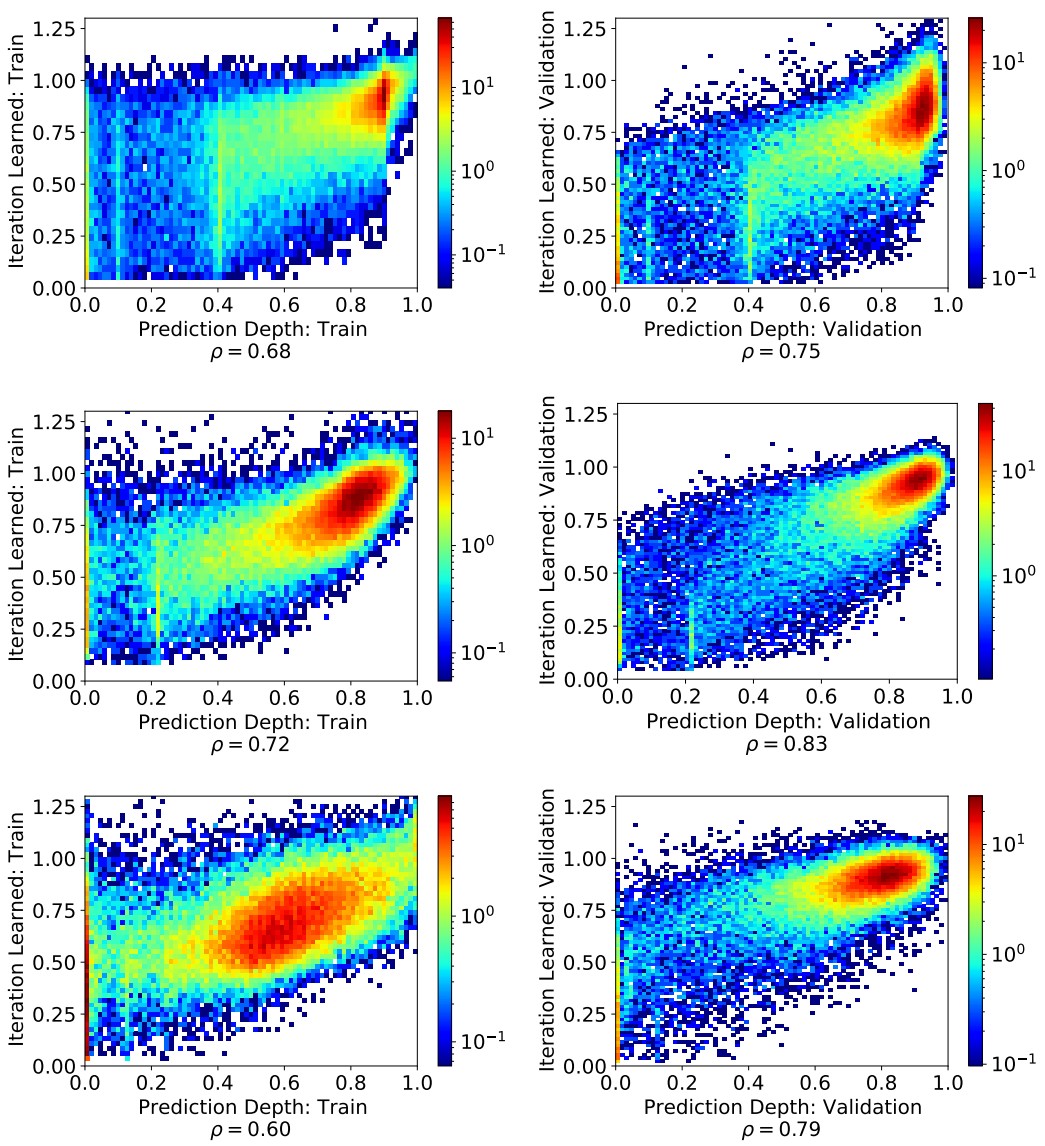

Figure C.25: CIFAR100. Top row: ResNet18. Middle row: VGG16. Bottom row: MLP. Histogram comparing the mean prediction depth to the mean iteration learned when each data point occurs in either the training split (left column) or the validation split (right column). See Appendix C.4 for a description of the experiments performed.

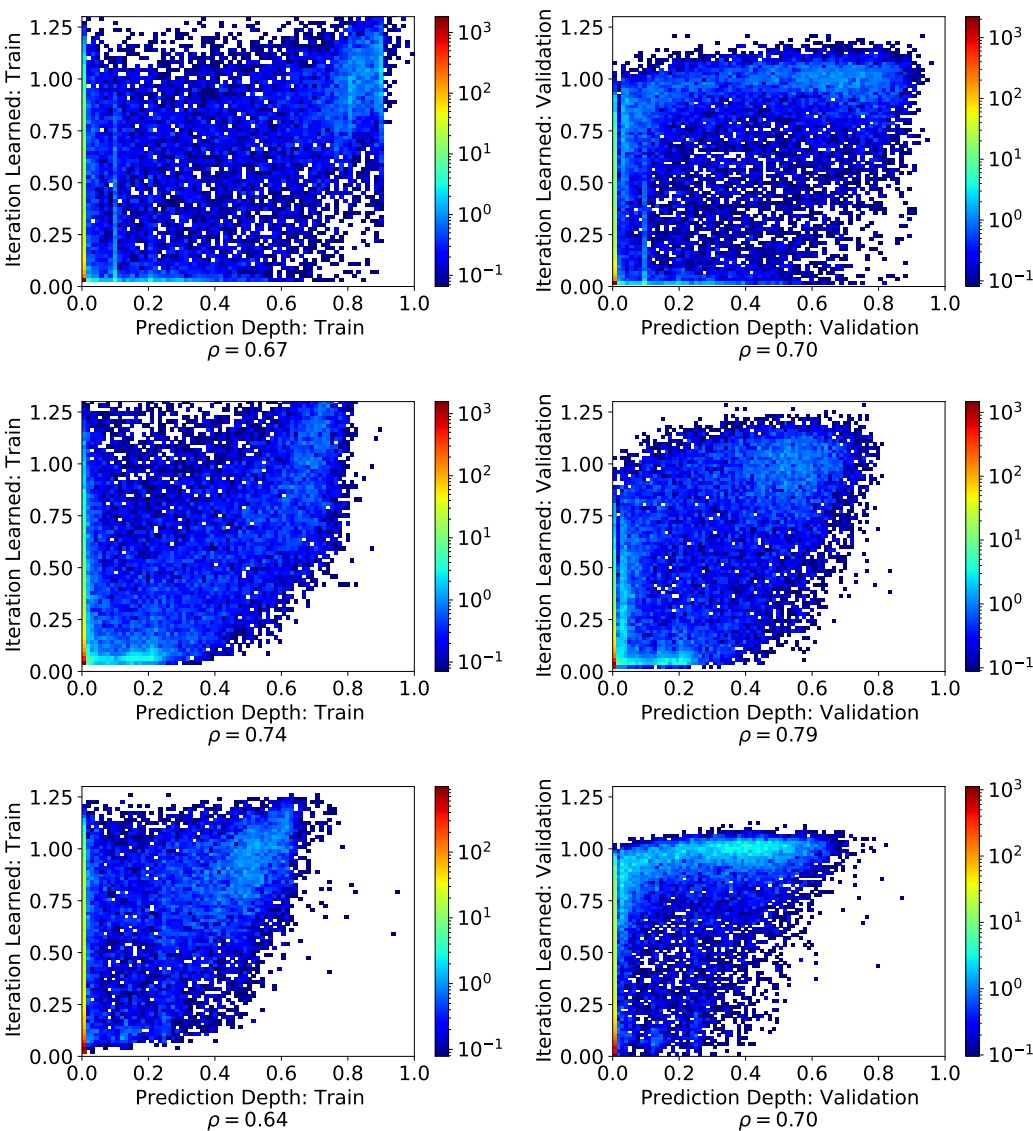

Figure C.26: Fashion MNIST. Top row: ResNet18. Middle row: VGG16. Bottom row: MLP. Histogram comparing the mean prediction depth to the mean iteration learned when each data point occurs in either the training split (left column) or the validation split (right column). In this case, the large majority of the data is already learned in the input layer. See Appendix C.4 for a description of the experiments performed.

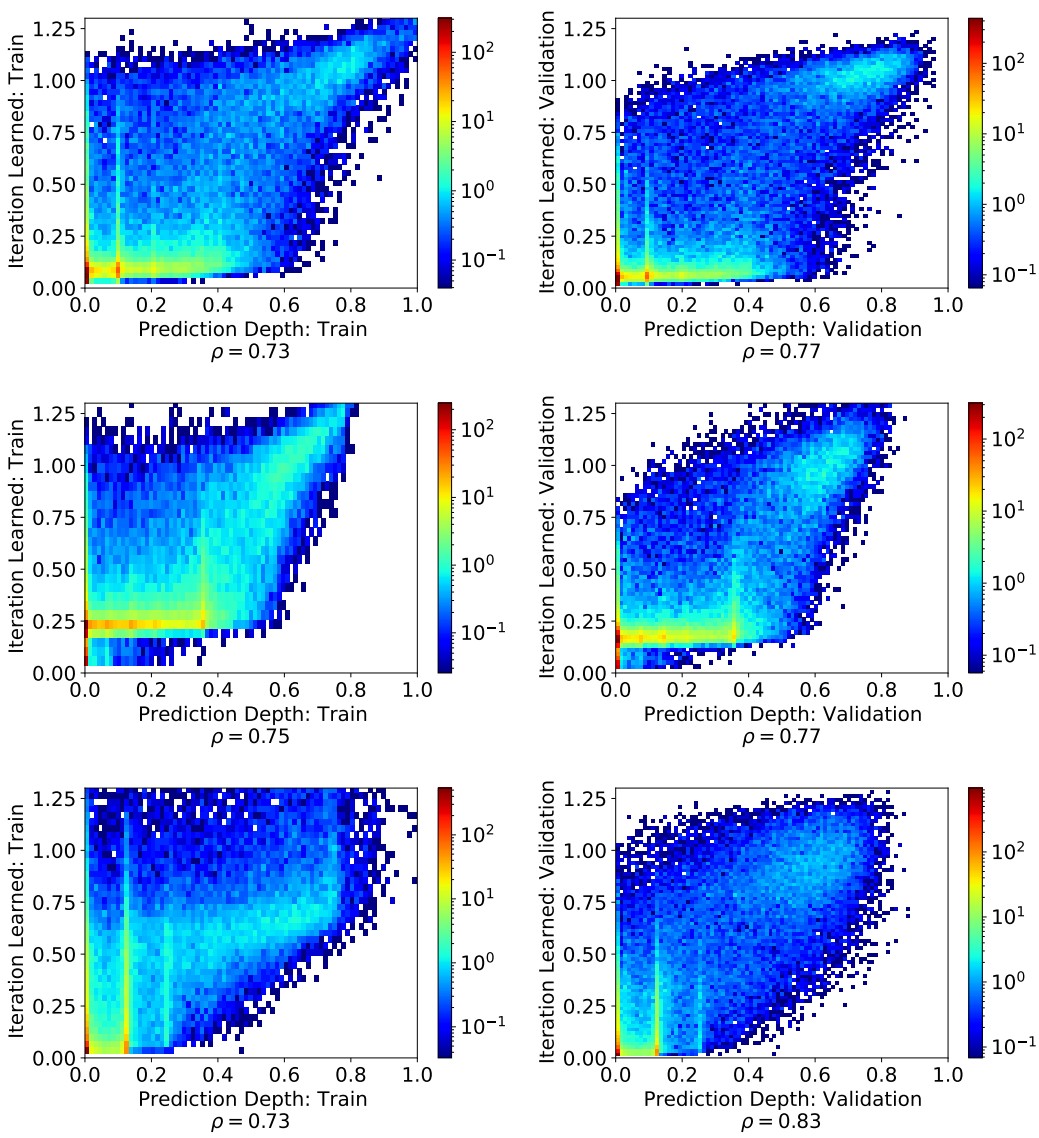

Figure C.27: SVHN. Top row: ResNet18. Middle row: VGG16. Bottom row: MLP. Histogram comparing the mean prediction depth to the mean iteration learned when each data point occurs in either the training split (left column) or the validation split (right column). See Appendix C.4 for a description of the experiments performed.

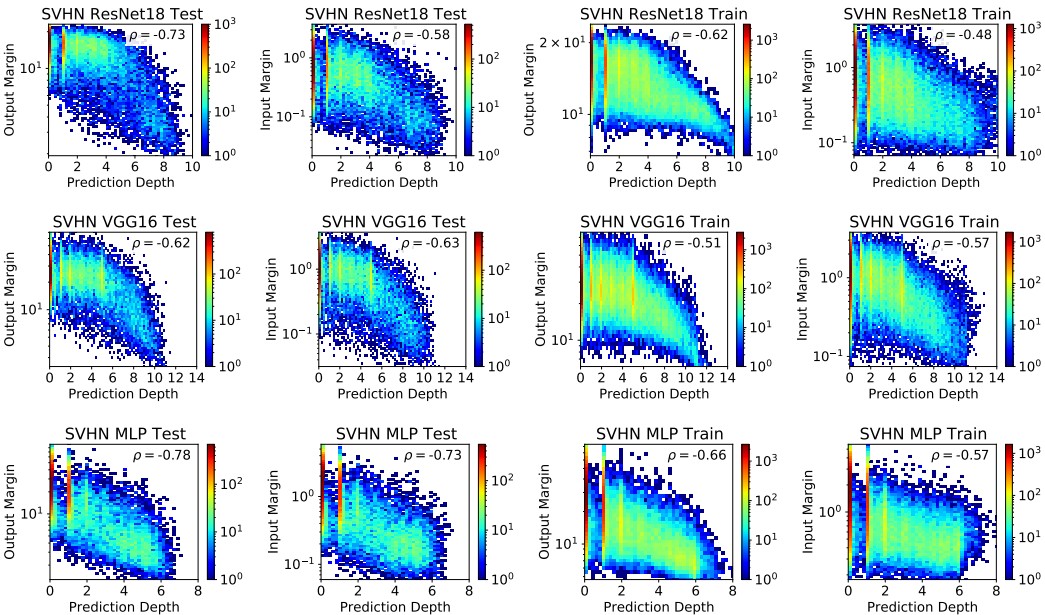

Figure C.28: Consistency of Figure 6, showing the correlation between prediction depth, and the input and output margins (log scale) for both the test and training splits of SVHN. The correlation coefficient between the prediction depth and the logarithm of the margin is given in each plot. For each architecture, we train 25 models with different random seeds on the full training split. We record the input and output margins together with the prediction depth for every data point in both the train and test splits. These histograms compare the mean values of each margin to the mean prediction depth for all data points.

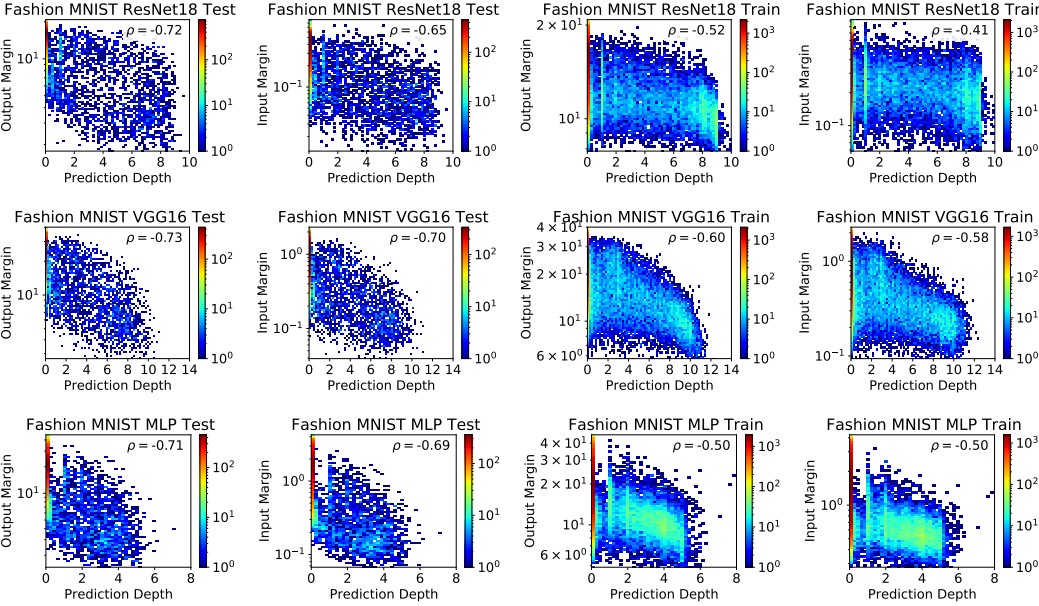

Figure C.29: Consistency of Figure 6, showing the correlation between prediction depth, and the input and output margins (log scale) for both the test and training splits of Fashion MNIST. The correlation coefficient between the prediction depth and the logarithm of the margin is given in each plot. For each architecture, we train 25 models with different random seeds on the full training split. We record the input and output margins together with the prediction depth for every data point in both the train and test splits. These histograms compare the mean values of each margin to the mean prediction depth for all data points.

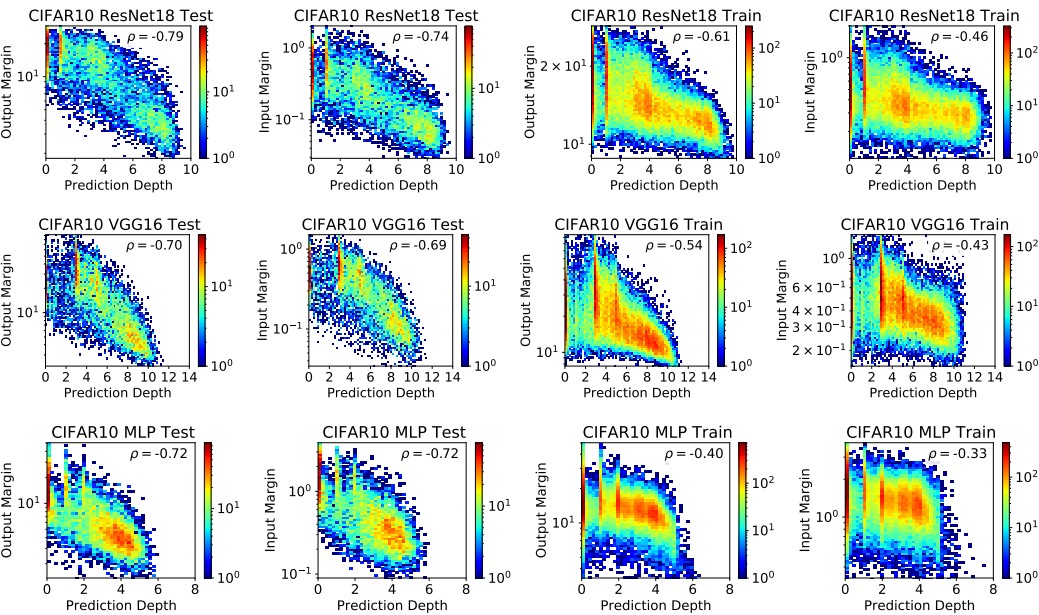

Figure C.30: Consistency of Figure 6, showing the correlation between prediction depth, and the input and output margins (log scale) for both the test and training splits of CIFAR10. The correlation coefficient between the prediction depth and the logarithm of the margin is given in each plot. For each architecture, we train 25 models with different random seeds on the full training split. We record the input and output margins together with the prediction depth for every data point in both the train and test splits. These histograms compare the mean values of each margin to the mean prediction depth for all data points.

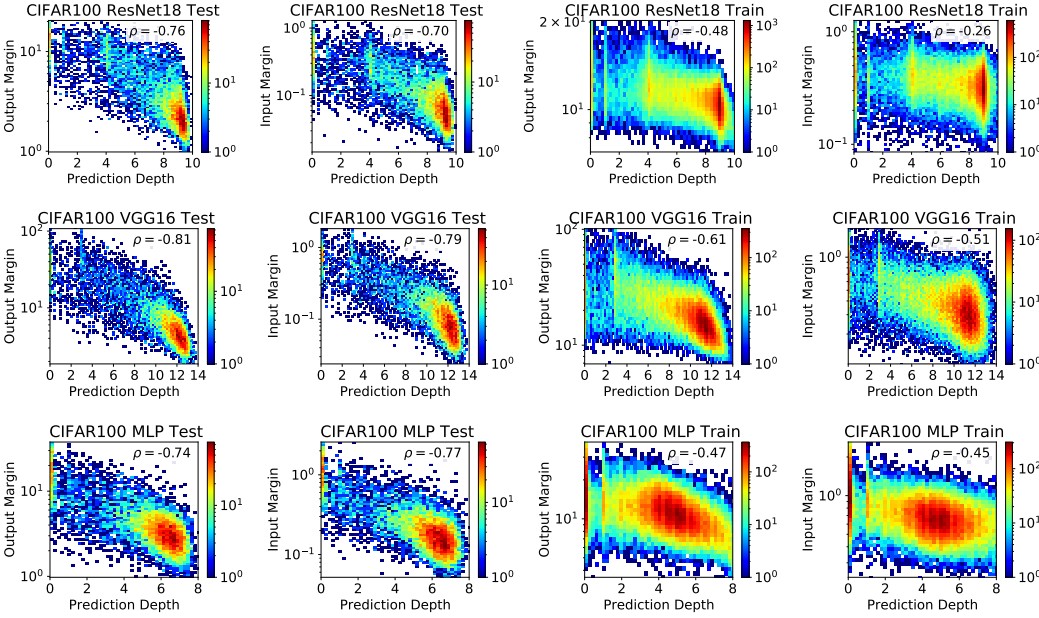

Figure C.31: Consistency of Figure 6, showing the correlation between prediction depth, and the input and output margins (log scale) for both the test and training splits of CIFAR100. The correlation coefficient between the prediction depth and the logarithm of the margin is given in each plot. For each architecture, we train 25 models with different random seeds on the full training split. We record the input and output margins together with the prediction depth for every data point in both the train and test splits. These histograms compare the mean values of each margin to the mean prediction depth for all data points.

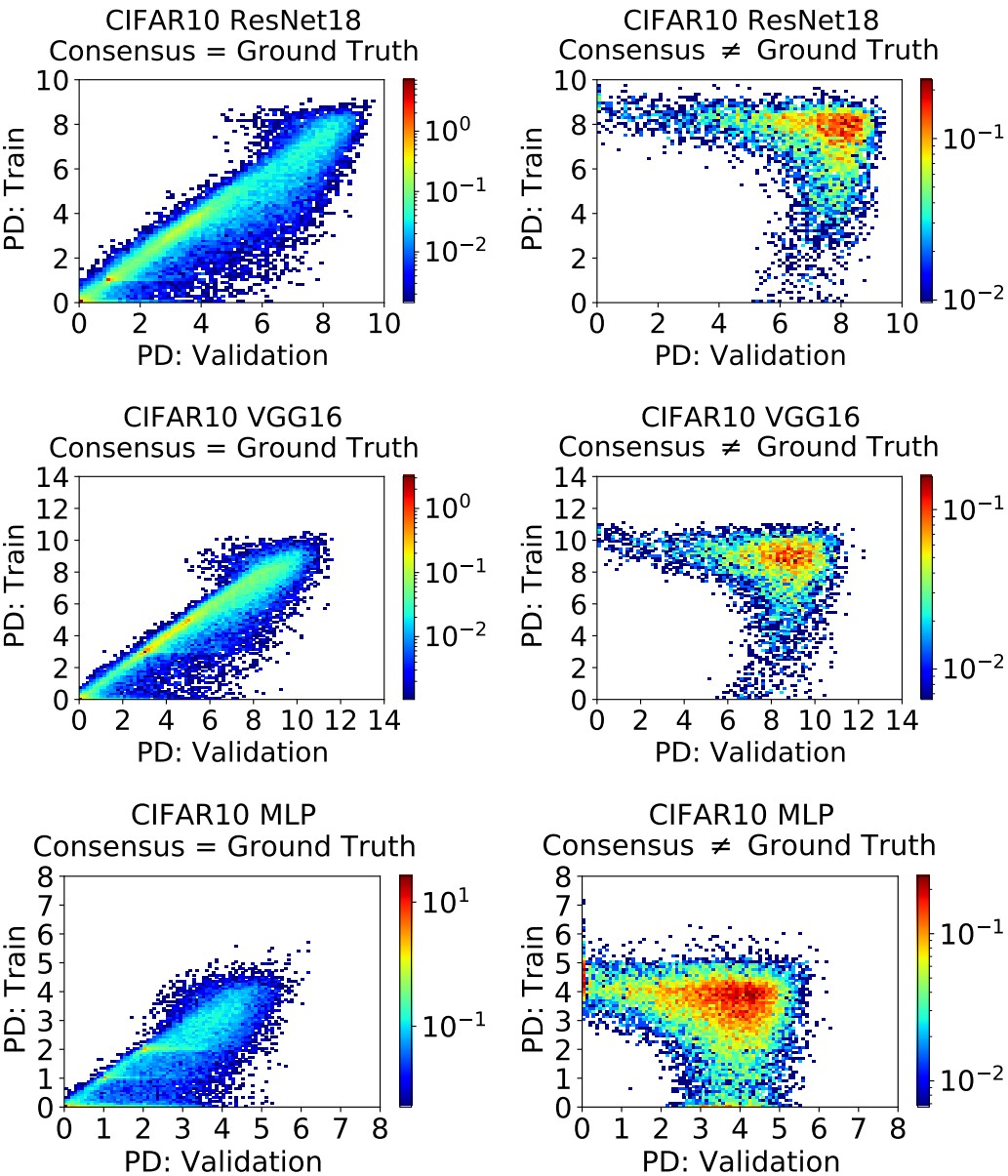

Figure C.32: Demonstrating consistency of the histograms shown in Figure 7 for all architectures on CIFAR10. These histograms compare the mean prediction depth when each data point occurs in either the validation split or the training split. Results are shown separately for data points where the consensus class is the same as or different from the ground truth label. See Appendix C.6 for a description.

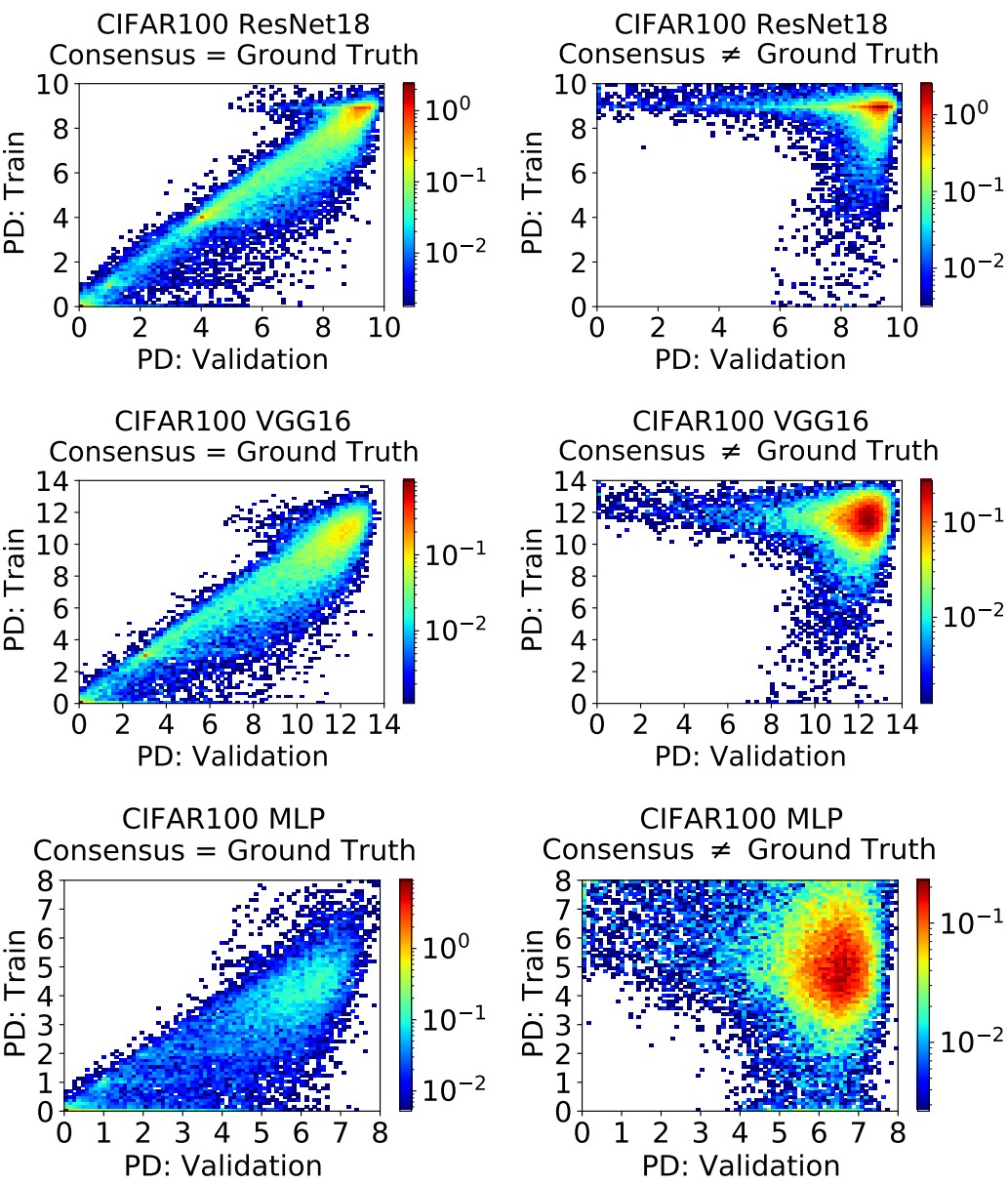

Figure C.33: Demonstrating consistency of the histograms shown in Figure 7 for all architectures on CIFAR100. These histograms compare the mean prediction depth when each data point occurs in either the validation split or the training split. Results are shown separately for data points where the consensus class is the same as or different from the ground truth label. See Appendix C.6 for a description.

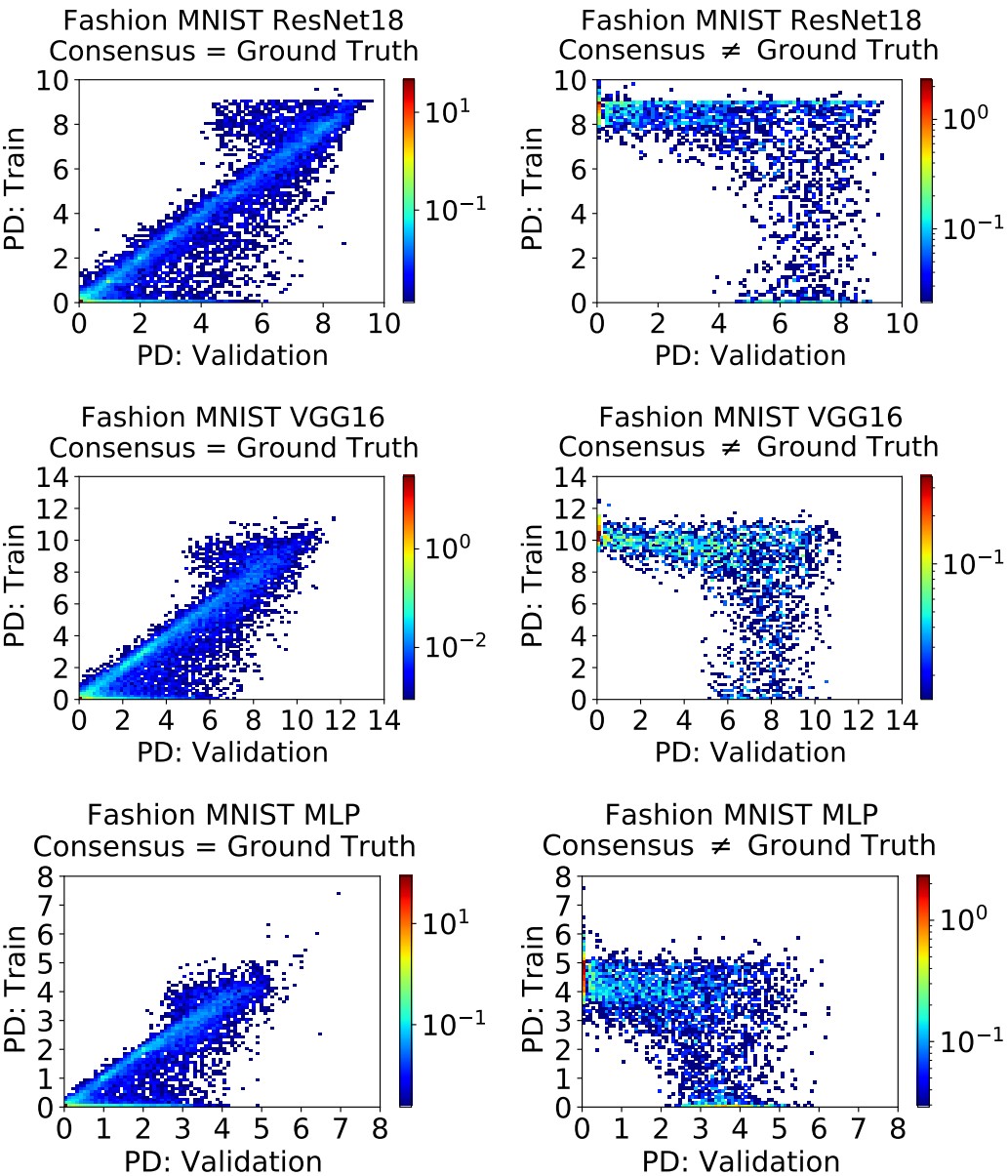

Figure C.34: Demonstrating consistency of the histograms shown in Figure 7 for all architectures on Fashion MNIST. These histograms compare the mean prediction depth when each data point occurs in either the validation split or the training split. Results are shown separately for data points where the consensus class is the same as or different from the ground truth label. See Appendix C.6 for a description.

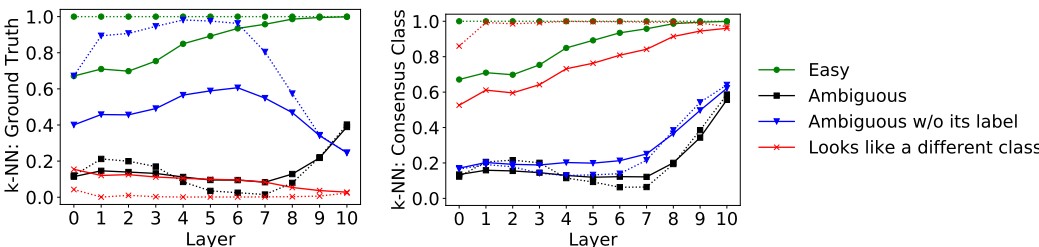

Figure C.35: Demonstrating consistency of the histograms shown in Figure 7 for all architectures on SVHN. These histograms compare the mean prediction depth when each data point occurs in either the validation split or the training split. Results are shown separately for data points where the consensus class is the same as or different from the ground truth label. See Appendix C.6 for a description.

Figure C.36: Reproducing Figure 8 for ResNet18 on CIFAR10.

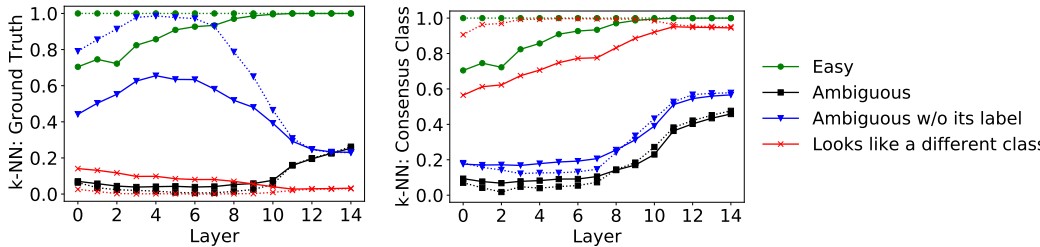

Figure C.37: Reproducing Figure 8 for VGG16 on CIFAR10.

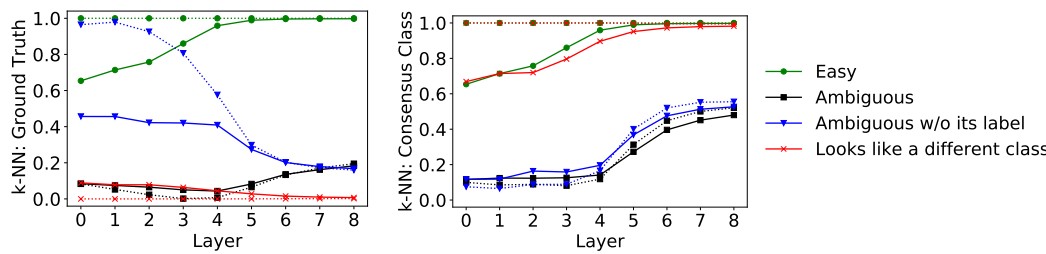

Figure C.38: Reproducing Figure 8 for MLP on CIFAR10.

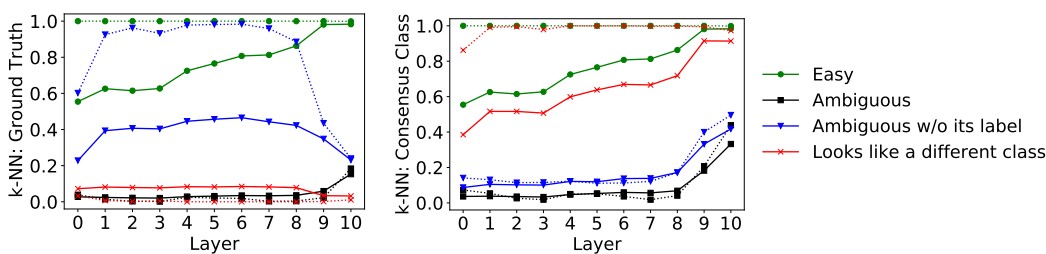

Figure C.39: Reproducing Figure 8 for ResNet18 on CIFAR100.

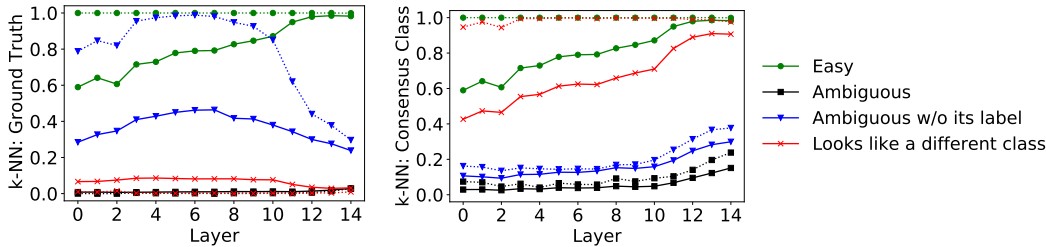

Figure C.40: Reproducing Figure 8 for VGG16 on CIFAR100.

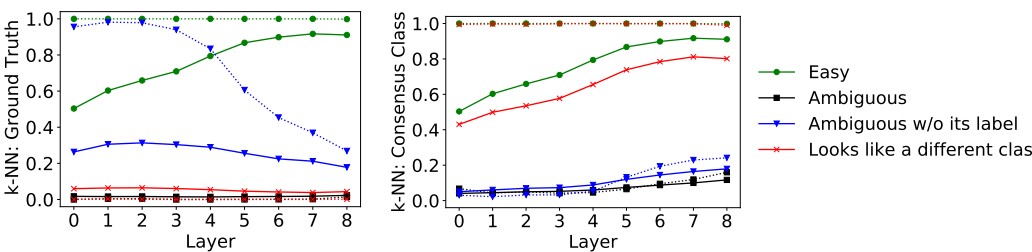

Figure C.41: Reproducing Figure 8 for MLP on CIFAR100.

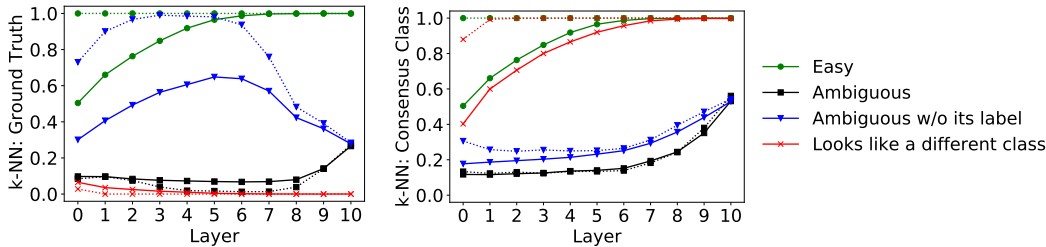

Figure C.42: Reproducing Figure 8 for ResNet18 on SVHN.

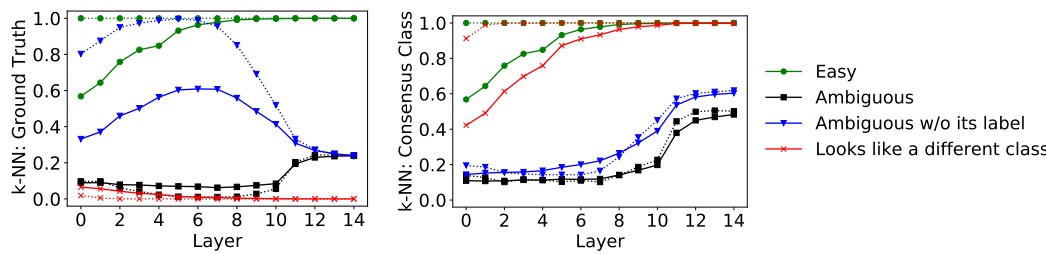

Figure C.43: Reproducing Figure 8 for VGG16 on SVHN.

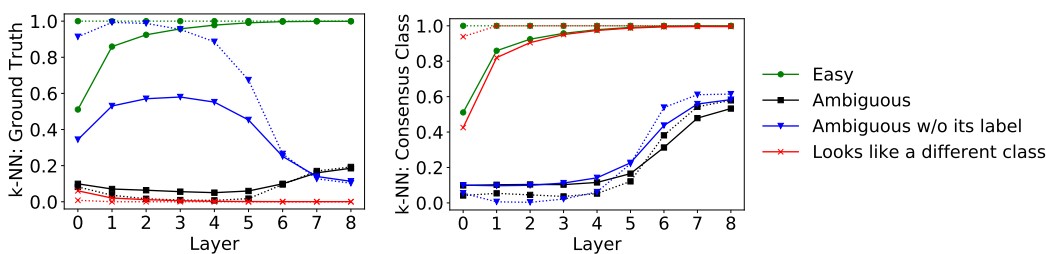

Figure C.44: Reproducing Figure 8 for MLP on SVHN.

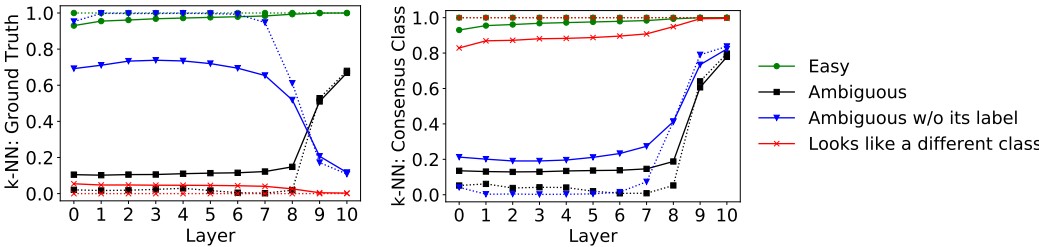

Figure C.45: Reproducing Figure 8 for ResNet18 on Fashion MNIST.

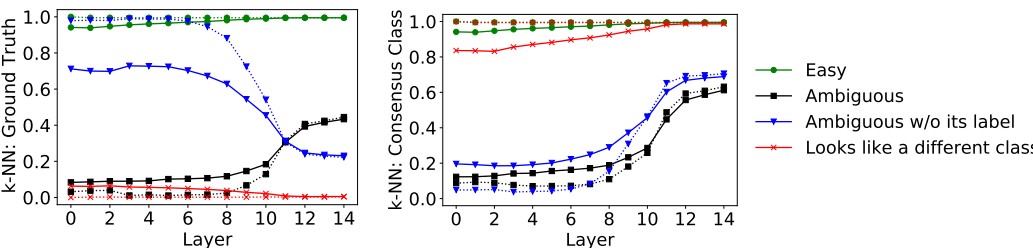

Figure C.46: Reproducing Figure 8 for VGG16 on Fashion MNIST.

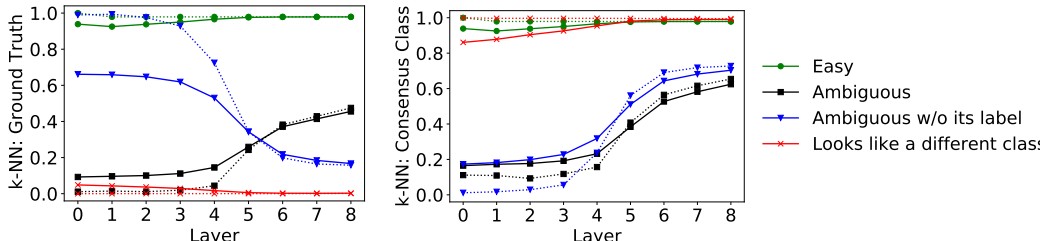

Figure C.47: Reproducing Figure 8 for MLP on Fashion MNIST.

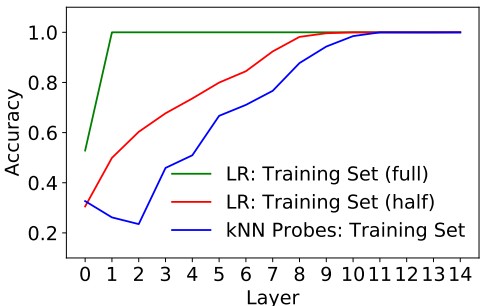

Figure E.48: Comparison of k-NN probe and Logistic Regression (LR) probe accuracies for VGG16 trained on CIFAR10. LR is already able to divide the training set into linearly separated classes after the first convolutional operation. In red we show the accuracy of LR probes trained on a random subset (half) of the data and predicting on the other half. These results are converged (closely repeatable between different trained VGG16 models).