# OpenReview forum: "Deep Learning Through the Lens of Example Difficulty"
_NeurIPS.cc/2021/Conference — NeurIPS 2021 Poster_

### Official Review · Reviewer_Cip6 · 2021-07-16

**Rating:** 5
**Confidence:** 4

**Summary:**

The authors proposed a new measure of computational difficulty called effective prediction depth, which determines the number of hidden layers after which the network's final prediction is determined by a k-NN classifier. They investigated different vision datasets and showed a consistency with a few known phenomena in the literature.


**Limitations And Societal Impact:**

As in the main review.

**Main Review:**

The understanding of deep learning models is an important field of research. Despite the good motivation, this work offers very limited novelty. Zhang et al., 2020 already proposed a way to evaluate the structural regularities of the labeled data. The observations made in the paper are also previously reported without significant new insights ("early layers generalize while later layers memorize; early layers converge faster; and networks learn easy data and simple patterns first").

The paper suffered from a major argument circularity issue. By using a KNN classifier, for each input data example, the "example difficulty" is determined mostly by the distribution of the training data (i.e. what are the other data examples). Thus, it shouldn't be a measure specific to a certain data example. For instance, if there is only one class in the dataset, does it mean that all examples have a prediction length of zero?

As another example, if the dataset is two-class that is highly imbalanced (0.1 and 0.9). Then, the large prediction length of the minority class shouldn't imply that it is "visually confusing or mislabelled" as suggested by the authors. It is just a data distribution artifact.

This measure is also highly dependent on the model architecture. If we only have 1-layer neural network, then all the examples will have a prediction length < 2. And the computed prediction length will have no meanings to compare to another models (say, a 100-layer network).

The observations made by the authors appear mostly obvious and reported by prior work. The KNN method to probe deep embeddings of each layer, as also pointed out by the authors, are proposed by Cohen et al. (2018) and thus, not novel.

-----
post-rebuttal:

As in the response to the rebuttal, I increased my score to borderline because I am more convinced on the insights that the prediction depth can give. The reason I wasn't able to raise the score further, is that, many of these insights (contextual property and unifying framework) can potentially require considerable writing improvements to emphasis their theoretical significance and interpretations, and without the option to review the revision, I can't fully root for this submission whose main contributions depends on these additional understandings.

**Time Spent Reviewing:**

5

---

> ### Author Response · Authors · 2021-08-11
> **Clarifying a key point: example difficulty is contextual, not absolute. It depends on the task, the training data and the implicit bias of the algorithm.**
>
> We thank Reviewer Cip6 for their time reviewing our work.
>
> Reviewer Cip6 cites Zhang et al., 2020. A key message of that paper is that the accuracy of a model for a given validation input (a measure of the input’s example difficulty) is conditional on the size of the training split. Reviewer Cip6 may not have understood this key message, because the rest of their review presupposes that example difficulty is absolute and does not depend on the other data in the training split, or the inductive bias of the algorithm, or even the task. We will explicitly highlight the contextual nature of example difficulty in the paper to make sure this is very clear. The contextual nature of example difficulty is well understood in the literature on example difficulty and we had thought that this was clear since the other reviewers have all understood this point.
>
> We will now respond to Reviewer Cip6’s remarks:
>
>
> Remark: “For instance, if there is only one class in the dataset, does it mean that all examples have a prediction length of zero?”
>
> Reply: Yes, if the task is to return the same label for every input then the task is trivial. This would be reflected by the prediction depth: the support set would be made up exclusively of examples with the same label in every layer and the prediction depth would be zero. This is exactly what we would want from a well-behaved measure of example difficulty.
>
>
> Remark: “...if the dataset is two-class that is highly imbalanced (0.1 and 0.9). Then, the large prediction length of the minority class shouldn't imply that it is "visually confusing or mislabelled" as suggested by the authors. It is just a data distribution artifact.”
>
> Reply: There is no contradiction here. Since the support set contains more inputs from the major class, the (majority vote) k-NN probes will, with high probability, only predict the minor class in layers where the activations for training inputs in the minor class have become sufficiently separated from those of the major class. Without class-identifying shortcuts, this could be expected to require more layers on average than for the major class. This reflects the intuitive notion that it should be harder to predict the output in classes with less training data. It is what we would expect from a well-behaved notion of example difficulty.
>
>
> Remark: “If we only have 1-layer neural network, then all the examples will have a prediction length < 2. And the computed prediction length will have no meanings to compare to another models (say, a 100-layer network).”
>
> Reply: This paper is concerned with deep learning. We haven’t worked on 1-layer networks. However in future work it would be interesting to compare what the prediction depth does tell us for successively shallower networks.
>
>
> We give references for the phenomena "early layers generalize while later layers memorize; early layers converge faster; and networks learn easy data and simple patterns first". One of our contributions is to connect these phenomena in a single experimental setup and viewpoint. To our knowledge, this is new.
>
> We reference Cohen et al. (2018) in the paper. They do not introduce the prediction depth or report the results that we report.
> Thank you for your time reading our manuscript. We hope that we have answered your key questions about the paper. If so, we would be pleased if you could improve your score to reflect this. We will stress the contextual nature of example difficulty.

---

> > ### Comment · Reviewer_Cip6 · 2021-08-19
> > **Thanks for clarification; Increase the score as some concerns addressed; Borderline.**
> >
> > First, I would like to thank the authors for the helpful clarifications on my concerns. After reading other reviewers' feedback and your responses, I believe some of my original concerns are addressed.
> >
> > On the contextual property of the example difficulty. It is important to highlight the contextual nature of example difficulty, because the context poses a sampling effect on the conclusions one might draw from the prediction depth. By categorizing the examples based on the high and low prediction depths, the context can have an effect on what the classified examples are distributed along the consistency score dimension. It would be beneficial to elaborate on these effects in the next version of this work.
> >
> > On the unifying perspective. The response reads, "one of our contributions is to connect these phenomena in a single experimental setup and viewpoint." I agree with the authors that this framework can be potentially insightful if they can be further elaborated.
> >
> > Other than these two comments, I also agree with the Reviewer SfKb's evaluation that "I think a paper focusing on these two contributions would present a valuable addition to the field. At the moment, however, the paper structure doesn't study these two phenomena in enough depth that I'm fully satisfied with its contribution on these two points alone. The current narrative focus of the paper on more generic notions of example difficulty is significantly less novel compared to the previously mentioned contributions, as it largely shows that this measure correlates with other measures already largely agreed to measure difficulty and didn’t seem to provide much information gain over what these other measures have already shown." Although the two contributions that I found most interesting is not the same as mentioned in the above quote, but I share a similar evaluation that that they would require further conceptual depth to make this a strong or successful submission.
> >
> > Given these updates, I increased my score to borderline because I am more convinced on the insights that the prediction depth can give. The reason I wasn't able to raise the score further, is that, many of these insights (contextual property and unifying framework) can potentially require considerable writing improvements to emphasis their theoretical significance and interpretations, and without the option to review the revision, I can't fully root for this submission whose main contributions depend on these additional understandings.

---

### Official Review · Reviewer_SfKb · 2021-07-16

**Rating:** 5
**Confidence:** 4

**Summary:**

Summary: this paper presents a measure of the difficulty of classifying an input example in a dataset defined by the layer of a neural network based on how a k-nearest-neighbours classifier clusters the input point at each layer of the network. The authors empirically evaluate the correlation between this measure and other notions of example difficulty such as margin width.

**Limitations And Societal Impact:**

Yes, theoretical work.

**Main Review:**

Strengths:
- The main idea, that network representations cluster some data points by earlier layers than others and that earlier clustering occurs for examples that the network learns faster and more reliably, is soundly supported by the empirical evidence provided in the paper.

- The paper includes a number of nice visualiizations of various properties of the prediction depth measure.

- Most experiment descriptions were reasonably clear.

- Tracking how networks cluster their inputs at different layers sounds like an intriguing concept that might be of interest in the neural network interpretability setting, though that is not where my expertise lies.

- The measure proposed does indeed seem to correlate reasonably well with a variety of other notions of example difficulty, and the authors have done a nice job of studying its correlation with a number of existing phenomena.

- The strength of some of the correlations between the accuracy of a network is striking, particularly the sharp triangular shapes shown in Figures 3 and 4.

Weaknesses:
- While the paper presents some interesting observations, these observations do not lead to actionable insights. The results here are observational, showing that a phenomenon appears in a set of architectures and datasets, but not making testable predictions about how networks should be trained in light of these observations.

- The results are not surprising. It is already known that intermediate layers perform clustering their inputs, and similar insights have been used in the uncertainty quantification literature. The fact that some datapoints are assigned to their cluster earlier in the network than others is not surprising given previous work studying early vs later layers of learning.

- The unifying picture starting in line 274 that connects prediction depth to previously observed phenomena doesn’t draw a clear causal connection between example difficulty and the related points. I mostly agree (though I think it might be worth explaining why consistent predictions imply generalization) that the observations on predictive depth are consistent with prior work, but it’s not clear to me whether the authors are suggesting that networks converge from input layer towards output layer because of some mechanism that depends on prediction depth, or whether they’re simply observing a correlation. For example, if networks converge from early to late layers and easy examples are learned first, then we would expect that the easy examples will be learned by early layers, and so will have low prediction depth.

- I don't see what additional information Prediction Depth provides about example difficulty beyond what would be obtained by existing methods such as looking at the number of training steps required to learn the example.


Concrete points for improvement:

- The “intervention” experiment is not clear. A great number of details of the training procedure are changed, and I don’t understand how the authors make the claim that they have isolated the relationship between margin width and prediction depth.

- The main point missing from this paper is an example showing that prediction depth is able to provide useful and/or novel insights into training and inference in deep neural networks. For example, the introduction suggests that formalizing principles governing the passage of data through layers of a neural network could help with curriculum learning, fairness, efficiency improvements in neural networks, uncertainty quantification, and distribution shift. I would be much more convinced that prediction depth is a useful and interesting concept if the authors showed an example where other notions of example difficulty fail to provide useful insights into one of these areas while prediction depth succeeds.

- Can the authors provide a clearer sense of whether they think that the phenomenon that prediction depth measures is a causal factor in the other deep learning phenomena they describe? In the absence of insights leading to algorithmic improvements, there may be interesting theoretical results that can be derived from the perspective of clustering and information propagation through layers. Providing such a unifying view would significantly strengthen the paper.


Other questions:
- The proposed measure depends heavily on the training data set. I would have liked to see a deeper analysis of how the training data set size and complexity affects the example difficulty of various inputs. For example does the difficulty of a given example change when the network is trained on CIFAR-10 vs CIFAR-100?

- How does the example difficulty of an input change over the course of training? Do networks trained to overfit vs with early stopping exhibit different behaviour?

**Time Spent Reviewing:**

4

---

> ### Author Response · Authors · 2021-08-11
> **Actionable insight; prediction depth is not redundant; scientific approach; intervention in Sec. 3.3; surprising results.**
>
> We would like to thank Reviewer SfKb for their careful reading of our work and the time taken to review our paper. We aim to satisfy their questions here.
>
> Actionable insight:
> In Figure 8, reporting the prediction of the k-NN probe in layer 4 for examples that are most “Ambiguous w/o their labels” would starkly increase the accuracy of those predictions. For the 100 most extreme examples of this form of difficulty, the accuracy of the probe in layer 4 is 98%, while the network has an overall accuracy of 25% for these examples. You could identify examples for which to report the k-NN probe’s class with those for which the k-NN probe in layer 4 has high confidence in a first class, and the prediction of the full network is different. This is an interpretable and actionable insight and it is derived from the prediction depth since that is how we determine the forms of example difficulty in Figure 7. We would be happy to describe this in the text.
>
> “Prediction depth is related to other measures of example difficulty. Doesn’t this make it redundant?”:
> If a measure of example difficulty were to be discarded, then we contend that you would rather prefer to keep the prediction depth, which is directly connected to the internal processing of data in the network and thus informative of the mechanics inside a deep model, as opposed to say, iteration learned, which is an “external” observable that treats the model as a black box.
>
> Causal mechanisms of deep learning:
> Rather than directly examining causal graphs between the different measures of example difficulty (which are hard to access because these concepts are relatively high level), we seek to demonstrate that the correlations shown in the paper are predictive, not spurious. We do this by making and testing predictions based on the observed correlations. In section 3.2 we predict, based on the correlation between iteration learned and prediction depth, that there should be a visual correspondence between the training and inference learning curves, and observe this to be true for important features. In section 3.3 we go further: based on the observed correlation between average prediction depth and average output margin, we predict that an intervention we design should both reduce the output margin and increase the prediction depth. We find this prediction to hold true.
> This is in essence, the scientific method, which was devised for scenarios where it is infeasible to reduce all behaviors to the smallest interacting quantities. That is exactly the scenario we find ourselves in when studying deep learning in terms of relatively high-level notions like example difficulty. We hope this clarifies our approach.
>
> Clarity of the intervention in Section 3.3:
> We don’t claim to isolate the relationship between margin width and prediction depth: we are following the approach described in the preceding paragraph. However, to ensure that the intervention is meaningfully connected to the control experiment we make a series of four experiments and show that there is a spectrum of behavior between the intervention and the control. This is described in Appendix A.7.
>
> Results are not surprising:
> We believe that the neat linear bounds shown in Figures 3 and 4 are highly surprising. The ability to observe how example difficulty separates into different forms, according to a deep model, is also a surprising advance and surely useful. We are also not aware of another work that connects these separate aspects of deep learning in one viewpoint and experimental setup.
>
> It would be good to study how example difficulty is conditional:
> We agree, this is a nice idea for future work. For example, in CIFAR10, birds against a blue sky have strong “Looks like a different class”, being mistaken for airplanes, while airplanes against a blue sky are “Easy”. If you changed the ratio of these examples you would expect the two to exchange their forms of difficulty at a certain point. Birds against a blue background are shown to be mistaken for airplanes in Figure 7.
>
> We thank you for your careful review. If you are satisfied with our response we would ask you to please increase your score.

---

> > ### Comment · Reviewer_SfKb · 2021-08-12
> > **Increasing confidence in empirical significance, but  not fully convinced to increase score**
> >
> > I’d like to thank the authors for their thoughtful response to my questions and comments. I agree that Figure 3 and 4 are striking, and although I had previously been skeptical of the apparent upper bound on the consistency measure that seemed to perfectly correlate with prediction depth, I now see in the appendix that a similar phenomenon seems to hold with different “slopes” for a variety of architectures and datasets. This property of prediction depth seems worthy of further investigation: I am surprised that each layer of the network seems to contribute equally to the change in the consistency measure, rather than earlier layers contributing more. I would be interested in the authors’ take on whether this is an innate property of neural networks (i.e. each layer extracts the same amount of information from the data) or whether this is a property more specific to how the k-NN classifier works. I think a principled justification for why we should expect to see these empirical results is currently missing from the paper, and would significantly strengthen these results.
> >
> > The categorization of different examples based on high/low PD on the train/test set is interesting. I actually find this a much stronger justification for preferring prediction depth over other measures of example difficulty than the justification given in the authors’ response: that PD is  “…directly connected to the internal processing of data in the network and thus informative of the mechanics inside a deep model, as opposed to say, iteration learned, which is an “external” observable that treats the model as a black box.” I don’t think measures like iteration learned or variance of gradients can distinguish how easy an example is to learn vs how easy it is to classify in the test set, and this seems like a really interesting property of the data to study further. Indeed, this section seems to leave an obvious question unanswered: what distinguishes the “Ambiguous unless the label is given” images from the “ambiguous” images? Does removing these data points from the dataset affect generalization?
> >
> > I think a paper focusing on these two contributions would present a valuable addition to the field. At the moment, however, the paper structure doesn't study these two phenomena in enough depth that I'm fully satisfied with its contribution on these two points alone. The current narrative focus of the paper on more generic notions of example difficulty is significantly less novel compared to the previously mentioned contributions, as it largely shows that this measure correlates with other measures already largely agreed to measure difficulty and didn’t seem to provide much information gain over what these other measures have already shown.
> >
> > While I appreciate the link to appendix A.7 and the corresponding ablation study, I don’t think it succeeds in justifying subsection 3.3’s title. In fact, the title suggests that by changing the training procedure, we have changed the example difficulty. Given that the dataset remains constant and only the training process changes, it seems a bit odd to suggest that this intervention has changed the example difficulty. I’m still not entirely sure what the experiment is showing aside from “changing the training procedure in a way that reduces margins also makes it harder for a k-NN classifier to distinguish datapoints”, which, relative to some of the other results in the paper, doesn’t seem like a particularly significant insight.
> >
> > The mini-summary of this update is that I’m more convinced that prediction depth is telling us something interesting about neural networks, but less convinced that the paper is effectively conveying this fact than I was when writing my original review. It’s a shame that NeurIPS isn’t letting authors upload revised PDFs on openreview, as I would be much more confident in raising my score if the authors were able to demonstrate improvements in the conceptual understanding of Figures 3 and 4, and potentially additional experimental evidence showing that the partition of the dataset into the 4 categories can provide insight into the effect of examples on network generalization and training.

---

> > > ### Author Response · Authors · 2021-08-22
> > > **Information and reported linear bounds as intrinsic properties of neural networks; Paper narrative; Removing data points from the training set.**
> > >
> > > We would like to thank Reviewer SfKb for taking the time to carefully read our response and re-evaluate our paper. In this second response, we aim to address your remaining concerns.
> > >
> > > **“I would be interested in the authors’ take on whether this is an innate property of neural networks (i.e. each layer extracts the same amount of information from the data) or whether this is a property more specific to how the k-NN classifier works. I think a principled justification for why we should expect to see these empirical results is currently missing from the paper, and would significantly strengthen these results.”**
> > >
> > > This is an interesting question. In Appendix Fig. C.18 we have shown that the mean prediction depth for an input gives an approximately linear upper bound on the negative information (entropy) of the predictions from the corresponding ensemble. We would be happy to make this result more prominent, referring to it at greater length in the main text.
> > >
> > > We anticipate that the linear bounds presented in Figures 3 and 4, using k-NN probes, are very likely an innate property of neural networks in the sense that they should also apply to the analysis of different architectures and different classification tasks. This is based upon their appearance for all four datasets with each of the MLP, ResNet and VGG16 architectures, which have different inductive biases. We also suspect that the bounds would be linear for other choices of probe (other than k-NN), however testing this remains for further investigation.
> > >
> > > **“The current narrative focus of the paper on more generic notions of example difficulty is significantly less novel compared to the previously mentioned contributions, as it largely shows that this measure correlates with other measures already largely agreed to measure difficulty and didn’t seem to provide much information gain over what these other measures have already shown.”**
> > >
> > > We believe that deep learning needs a predictive theory. Before a quantitative theory is built, it is helpful to establish coherent empirical viewpoints that are demonstrably able to describe diverse aspects of deep learning. Many of the aspects one would want to connect have naturally been studied before but, which is important, using disjoint experimental viewpoints.
> > > The paragraph “Connecting known phenomena” which starts on line 274 connects four separate deep learning phenomena to the prediction depth. This is a key contribution of the paper which we think is valuable. To connect these phenomena the paper must report the correlations in sections 3.2 and 3.3. These results, strictly speaking, are new. If such results are forbidden because they do not seem sufficiently surprising, then it will be very hard for any paper to demonstrate how a single viewpoint relates these diverse aspects of deep learning.
> > >
> > > As a secondary point, showing the relationships between prediction depth and these diverse measures of example difficulty strengthens the case for considering prediction depth as a measure of example difficulty.
> > > We are happy to improve the introduction and contributions section, to more clearly justify reporting the results in sections 3.2 and 3.3. We are also happy to give the paragraph “Connecting known phenomena” on line 274 its own subsection. We hope that this explains why, in our view, it is important to report the correlations shown in sections 3.2 and 3.3.
> > >
> > > We are happy to change the title of Sec. 3.3 as follows, which we hope solves your objection to the title:
> > > “3.3 Deep models exhibit larger margins for inputs with lower prediction depth”
> > >
> > > **“What distinguishes the “Ambiguous unless the label is given” images from the “ambiguous” images? Does removing these data points from the dataset affect generalization?”**
> > >
> > > We will first address the difference between these two subsets of data points.
> > >
> > > Fig. 8 shows how, in the early layers of a network, images that are strongly “Ambiguous unless the label is given” have many neighbors in their ground-truth class. If these inputs were highly noisy in the pixel space then this would be unlikely to occur. Hence we can assume that such inputs are visually sensible. In early layers, they are processed as members of their ground truth classes, but in later layers they are processed as members of a different, consensus, class.
> > >
> > > In early layers, examples that are highly “Ambiguous” have few nearest neighbors in either their ground truth or consensus classes. In the datasets we consider, these inputs are characterized as both visually highly atypical examples, images taken from strange perspectives, and also inputs with very high input noise.
> > >
> > > Extreme inputs from both these sets (not hand-picked) are shown in Fig. 7 for CIFAR10 class “Birds”.
> > >
> > > You asked about the impact of removing these data points on generalization. The answer to this question depends strongly on the makeup of the dataset.
> > >
> > > In Feldman and Zhang (2020) the authors report that typical image classification datasets contain a “long tail” of images that are members of visually atypical subgroups. Such images might include ostriches and flamingos in the class “Birds”, but they can also refer to smaller groups such as a “Table” fashioned from a barrel. Not learning inputs from these subgroups disproportionately affects the generalization of the model on test images from the same subgroups.
> > >
> > > In our visual inspection (a little of which is shown in Fig. 7), we found that images that are members of rare subgroups may be either “Ambiguous” or “Ambiguous unless the label is given”. Removing these inputs from the training split will alter the generalization of the model differently for the “Easy” test inputs vs. test inputs that are not “Easy”. It depends on the makeup of the data whether this will improve or decrease average accuracy. We also anticipate that it would lead to a model with performance fall-offs between easy and difficult inputs that are larger than if you train on the whole data set, which could be expected to impact the ethical/fairness qualities of the model.
> > >
> > > We would be happy to describe this in more detail in the paper.
> > >
> > > If you feel that your questions have been addressed, we would ask you to please improve your score.
> > >
> > > [1] Feldman, Vitaly, and Chiyuan Zhang. "What Neural Networks Memorize and Why: Discovering the Long Tail via Influence Estimation." Advances in Neural Information Processing Systems 33 (2020): 2881-2891.

---

### Official Review · Reviewer_nMGW · 2021-07-18

**Rating:** 8
**Confidence:** 4

**Summary:**

The authors propose a new characterization of the difficulty of an example during training and testing of deep learning models: the layer at which an example is learned (which they call "prediction depth"). They show that early layers are enough to learn easy examples, and that difficult examples are primarily learned by later layers. They also connect the prediction depth to two established notions of example difficulty from the literature: the consistency score and the iteration learned (closely related to the example forgetting statistic).



**Limitations And Societal Impact:**

Yes

**Main Review:**

Strengths:
1. The authors present a comprehensive set of experiments to characterize the layer depth in deep neural networks at which examples can be classified correctly by a k-NN probe. They evaluate the prediction depth of examples from 4 datasets with varying difficulty (from fashion MNIST to CIFAR 100), and show that this prediction depth is relatively consistent across architectures and random seeds.

2. This manuscript also offer a helpful characterization of examples along two axes: the prediction depth of an example when it occurs in the training set and when the example occurs in the validation set. I'm specifically excited about this characterization's ability to disentangle "hard" examples into three groups. This characterization has the potential to inspire future work on better understanding these different groups and how to encourage learning that leads to better generalization.

Weaknesses:
1. The clarity can be improved. The paper is extremely dense, which is not helped by the obvious tampering with the NeurIPS template (e.g. there is basically no space between paragraphs). I appreciate that the authors wanted to include results from a thorough experimentation but the sheer amount of content can be counterproductive in getting their point across clearly. The authors also rely on the figure captions to summarize the figure takeaways without including summaries in the main text. I appreciate the descriptiveness of the figure captions but they often occur on different pages from where the figure is referenced, so having to continuously flip through the paper to just understand the takeaways of the experiment really breaks up the flow.

2. The connection between the findings in this work and applications of deep learning is entirely relegated to the Appendix. I recommend the authors to include a more thorough discussion of these possible connections in the main paper.

3. Lack of quantitative characterization of the claimed correlations in Figures 5-6. The authors merely state that the measured quantities in these figures correlate but do not report the actual values of the correlations. Also when the authors report a correlation coefficient, they should specify which correlation metric was used (Pearson?).

Minor points (mostly typos and clarity issues):
- L50 and L306: authors refer to "four deep learning phenomena" but only 3 are listed and appear in the paper
- Fig 2 refers to 250 trained models but at this point the authors have not discussed what these models are. The authors should also specify how many random seeds were used in Fig 2.
-  L120-L127: does the MLP achieve similar training and validation accuracy as the other two architectures? Can the differences in prediction depth be due to differences in these final accuracies?
- L163: "of in"
- What is actually plotted in the 2d histograms? Is it the percentage of the dataset that has specific properties?
- I didn't understand what understanding the intervention in section 3.3 really provides beyond the results already shown in Fig 6. I recommend the authors clarify this or remove it from the main paper, since they are short on space.
- Fig 8 caption: "extremes" -> "extreme"
- Fig 8: what are the std errors across these 100 examples?



**Time Spent Reviewing:**

5

---

> ### Author Response · Authors · 2021-08-11
> **Moving figures and Appendix D; Pearson's correlation coefficients; minor points.**
>
> We would like to thanks Reviewer nMGW for their time spent reading our work and in producing this careful and interesting review. We are also excited that it’s now possible to automatically separate these different forms of example difficulty, and look forward to insights and new approaches building on this work in the future.
>
> Weaknesses:
> 1. We can certainly move the figures to the correct pages. We considered it better to have one figure on each page, but we are happy to move the figures to improve readability.
> 2. If accepted, we would have an additional page of space, which would improve the readability of the paper and allow us to move discussion of the connections between this work and applications of deep learning into the main paper. We would be happy to do so.
> 3. The Pearson's correlation coefficient for Figure 5 (left) are as follows:
> CIFAR100: 0.83
> CIFAR10: 0.7
> Fashion MNIST: 0.79
> SVHN: 0.77
>
> In Figure 6 (left and middle) the Pearson's correlation coefficients between the log(margin) and the prediction depth are
> Output margin: -0.7
> Input margin: -0.69
>
> We are happy to add these numbers to the paper. For Figure 6 (left and middle), the correlation coefficients are additionally given for all combinations of dataset, architecture and split in Appendix C.5.
>
> Minor points:
>
> •	L50 and L306: we consider “networks learn easy data first” and “networks learn simple functions first” to be separately reported phenomena.
>
> •	We are very happy to clarify the models trained, and the number of random seeds, in Figure 2.
>
> •	L120-L127: All models are trained to convergence (zero training accuracy) but MLP does not achieve comparable validation accuracy to either ResNet18 or VGG16 (please see Table 2 in the Appendix for the accuracies of all models). It remains to be seen if the causal link is directly between the generalization of the model and the prediction depth, or if both are caused by some third unknown quantity, which might be the inductive bias of the model. The suspicion that the causal graph may contain a third node underlies what we have written in these lines.
>
> •	The 2d histograms show the probability density in bins on a 2d grid. The color bars are on a log scale. We are happy to write this explicitly in the text to make it clearer.
>
> •	In section 3.3 we form a hypothesis: “This intervention, which we expect to decrease the output margin, should also increase the prediction depth.” This hypothesis was formed based on the observed correlation between output margin and prediction depth. We interpret the fact that the prediction holds true as empirical evidence that the correlation between output margin and prediction depth is not spurious but connected to how deep learning works. We are happy to make this clearer in the paper.
>
> •	The std errors across these 100 examples vary with the layer and the form of difficulty. However, they are small enough that they do not distort the picture given. We prefer to leave them off because the figure is already rather complicated.
>
> We would like to thank you again for your thorough and engaging review.

---

### Official Review · Reviewer_GF8b · 2021-07-21

**Rating:** 6
**Confidence:** 3

**Summary:**

This paper introduces the prediction depth (PD) as a measure of example difficulty in deep networks. They then conduct an empirical study on several vision tasks of the relationship between PD and some other quantities:
 - difficulty of an example, as (qualitatively) measured by a human
 - accuracy and consistency
 - difficulty of learning an example (i.e. an example is difficult to learn if it is consistently correctly classified after a large number of iterations)
 - margin
and conclude that PD is useful for predicting these quantities

**Limitations And Societal Impact:**

yes

**Main Review:**

This work is in the line of other works trying to characterize example difficulty in deep learning. This is certainly a trendy topic, and an interesting one as far as I am concerned. The proposed measure of example difficulty gives new insights in how deep networks make their prediction. This new measure adds up to the growing list of example difficulty measure recently proposed in the literature, and I would be interested in seeing a comparative study of all different example difficulty measure (but that could be the subject of another paper).

The paper reads rather clearly, despite the impression that it is the shorten version of a longer paper. The experiments are clearly motivated, explained and discussed, except for section 3.3

The empirical relationship between PD and other quantities, observed in this case on vision tasks, is of significant interest.

During reading, I had the following questions:
1. Can you motivate the use of k-NN classifiers rather than anything else. What does it tell us about how a prediction is made, that something else e.g. a random forest would not tell.
2. in sec 3.3 I had a hard time understanding your experiment. "we construct a loss function that does not promote confidence" l221 => not only does it not promote confidence because of the hinge loss, the use of full batch gradient descent, and very small learning rates drastically change the dynamics of training. Aren't we hitting the lazy training regime of Chizat and Bach 2018? IMO the control experiment is too far from this setup (i.e. it uses large learning rates, and minibatch gradient descent with momentum).

I would recommend to accept it for NeurIPS, provided that these questions are satisfyingly answered.


**Time Spent Reviewing:**

4.5

---

> ### Author Response · Authors · 2021-08-11
> **Reasons for using k-NN; connection between 0-Hinge experiment in Section 3.3 and its control experiment.**
>
> We would like to thank Reviewer GF8b for their interesting questions and careful review of our paper.
>
> Query 1: Use of k-NN.
>
> Response: This paper seeks to better understand deep learning. For this reason we selected a model for the probes which makes interpretable predictions. In our view Random Forests are not interpretable in a simple sense. For example, the increasing accuracy (according to the consensus class of an input) of successive k-NN probes tells us that, to a zeroth-order approximation, deep models can be thought of as clustering inputs through successive embedding spaces from the input towards the output according to their consensus classes. In contrast, if we used Random Forests we would not have the same interpretable insight into the passage of data through the network.
>
> We did consider the use of other probes, and in particular logistic regression (LR). With the high dimensional embedding spaces in typical deep models like ResNet18, and for datasets with only ~10^4 data points, LR can easily separate the training split into linearly separable subspaces in very early layers. Fixes for this might include training multiple LR models on different random splits of the training set and then predicting on the rest, but then the prediction depth of an input would then depend on the hyperparameters used for training the LR probes, such as the size of the random split used to train LR. In contrast, Appendix Fig. A 10 shows that the results of k-NN probes are largely insensitive to the choice of k in the region of k = 30. Thus we chose k-NN over LR because the results are, in our view, less ambiguous for the training split than they would be with LR.
>
> Query 2: In Section 3.3 the control experiment seems too far from the new training method.
>
> We were concerned with the same question. This is why we additionally trained the two intermediate models shown in Appendix A.7. In total we report results for models trained in four scenarios:
> 1. Cross-Entropy + high learning rate SGD  (shown in the main paper)
> 2. Cross-Entropy + small learning rate gradient descent
> 3. 0-Hinge + high learning rate SGD
> 4. 0-Hinge + small learning rate gradient descent  (shown in the main paper)
>
> In Appendix A.7 we show that both interventions (changing the loss and changing the optimizer) are required to obtain a very small mean output margin (see Table 4 in the Appendix). There is a clear trend in the results from setups 1 → 4: average prediction depth gets larger and output margin smaller, following the sequence 1 → 2 → 3 → 4. Rather than being separate, scenarios 1 and 4 are at different ends of a spectrum, across which we observe a clear trend.
>
> If you are satisfied then we would be pleased if you would adjust your score. Thank you for your careful consideration of our manuscript.

---

> > ### Comment · Reviewer_GF8b · 2021-08-20
> > **Still not so convinced about the 2nd part of section 3.3**
> >
> > **LR vs k-NN**: Just a remark that 1-NN would also get a 0 training error regardless of the number of datapoints/embedding space size, similar to unregularized LR. That fact that you are using k-NN with k>1 is already a form of regularization, and regularized LR with proper hyperparameters (e.g. a ridge penalty) might do as well as a definition of PD, but arguably with a higher sensitivity to the choice of the regularization coefficient.
> >
> > **On the 2nd experiment in section 3.3**: I still think that the conclusion (or at least the conjecture) that you draw from the intervention experiment in section 3.3 seems rather shaky, given that other than the change in output margin, you are training your model in a different regime (small vs large learning rate, CE loss vs hinge). I don't think that this experiment provides enough insights in its current state.
> >
> > I also note that reviewers nMGW and SfKb expressed reserves regarding the clarity/significance of this experiment too.
> >
> > But overall I still feel that this paper is worth being published to NeurIPS, that is why I am still leaning positive.

---

### Decision · Program_Chairs · 2021-09-28

**Decision:**

Accept (Poster)

**Comment:**

The paper proposes Prediction Depth (PD) as a measure of example difficulty. The PD of an example is defined as the first layer for which the predicted label of a KNN classifier that uses the representations of the training set produced by that layer (and the KNN predictions for all subsequent layers) matches to the predicted label of the network. The papers finds that easy examples have low prediction depth (learnt early in the network). Differently from other measures of example difficulty, PD can be calculated for examples when they are included both in training set and in validation set. This allows the authors to characterize three types of example difficulty. They show PD is predictive of the consensus-consistency of a separately trained ensemble, which sheds potential applications to uncertainty estimation (though already done to some extent by [20] and [46]).

--

There have been extensive discussions amongst reviewers about this paper. This is an empirical paper. All reviewers agree that the topic is interesting and timely. The paper is generally clear although it appears too dense. A suggestion is to move some text from captions into the main text. Reviewers think that the experiments are comprehensive. The major concerns were around the a) novelty of some of the findings (low-prediction depth correspond to easy examples), b) the lack of actionable insights based on the observations and c) lack of conceptual depth (what does PD really tell us about generalization?). After rebuttal, reviewers agree that interesting findings and observations (characterization of train/val PD of an example, lower-bound on consistency-consensus score, eg Figure 3, 4, 7) are made.

Overall, this is a borderline paper. Due to the extensive amount of experimental results, I think this paper could potentially benefit the field in general. However, in the current form the paper might not be quite ready for publication yet and might benefit from an additional round of reviews. Following reviewers' advice, I encourage the authors to resubmit the paper by putting more emphasis on the findings in Fig. 3, 4 and 7, to discuss potential actionable insights of this work in the main text, to check their final copy for typos ("Abstract:" in the abstract, "LL" is not introduced anywhere in l249, ...) and to balance the overall textual load of the paper.

**Consistency Experiment:**

NeurIPS has a long history of experimentation. In 2014, NeurIPS ran an experiment in which 10% of submissions were reviewed by two independent committees to quantify the randomness in the review process. This year, we repeated a variant of this experiment to see how the quality of the review process has changed over time.  This paper was part of the experiment and was therefore assigned to two committees (consisting of reviewers, an Area Chair, and a Senior Area Chair) that reached independent decisions.  If both committees made the same recommendation, this recommendation was followed. If a single committee recommended acceptance, the paper was accepted (with the exception of a few cases in which the other committee identified what we considered a fatal flaw, e.g., an error in a key result).

This copy’s committee reached the following decision: **Reject**

The other committee assigned to the paper recommended **Accept (Poster)**.  You can find the other set of reviews, along with any follow up discussion with the authors here:
https://openreview.net/forum?id=fmgYOUahK9